# Semi-local bounds on null energy in QFT

**Jackson R. Fliss**[1*] **and Ben Freivogel**[1,2]

**1** Institute for Theoretical Physics Amsterdam,
**2** GRAPPA, University of Amsterdam,
904 Science Park, 1098 XH Amsterdam, the Netherlands

* j.r.fliss@uva.nl

## Abstract

We investigate whether the null energy, averaged over some region of spacetime, is bounded below in QFT. First, we use light-sheet quantization to prove a version of the "Smeared Null Energy Condition" (SNEC) proposed in [1], applicable for free and super-renormalizable QFT's equipped with a UV cutoff. Through an explicit construction of squeezed states, we show that the SNEC bound cannot be improved by smearing on a light-sheet alone. We propose that smearing the null energy over two null directions defines an operator that is bounded below and independent of the UV cutoff, in what we call the "Double-Smeared Null Energy Condition," or DSNEC. We indicate schematically how this bound behaves with respect to the smearing lengths and argue that the DSNEC displays a transition when the smearing lengths are comparable to the correlation length.



# 1 Introduction

Energy conditions play a distinguished role at the interface between classical and quantum physics. Nowhere, perhaps, is this better illustrated than in semi-classical gravity. Because every metric solves the Einstein equations for some choice of stress-energy tensor, energy conditions are needed to constrain the set of physically realizable spacetime geometries. Classically, one such constraint, obeyed by all sensible classical theories, is the Null Energy Condition (NEC),

$$\mathbf{T}_{\mu\nu}k^{\mu}k^{\nu} \geq 0, \qquad \text{for} \quad k_{\mu}k^{\mu} = 0, \tag{1}$$

where $\mathbf{T}_{\mu\nu}$ is the stress tensor of classical matter on a background geometry. Penrose showed, using the NEC as an assumption, that trapped surfaces must lead to singularities [2], ruling out traversable wormholes and bouncing cosmologies.

Quantum mechanically, the NEC is violated in even the most pedestrian of quantum field theories. Our world is quantum mechanical. Thus the pressing question of "What spacetime geometries can arise in semi-classical gravity?" requires further conditions on *quantum null energy*. Lacking such conditions, it is unclear whether trapped surfaces must lead to singularities and whether exotic spacetimes such as traversable wormholes and bouncing cosmologies can occur in semi-classical gravity.

Two important examples in this direction are the Achronal Average Null Energy Condition (AANEC) [3–7] and the Quantum Null Energy Condition (QNEC) [8–11]; see [12] for a nice review. While these results have varying degrees of applicability to semi-classical gravity, they also illuminate the fact that energy inequalities are interesting objects in their own right for a quantum field theory, revealing an interesting interplay between null energy, causality [13], and quantum information [7, 10].

While these previous results are of great interest, we note an important drawback to the quantum energy conditions mentioned above: namely, the constraints they impose on the null energy are either completely non-local (in the case of the AANEC) or state-dependent (in the case of the QNEC). The QNEC, in particular, is motivated by an elegant and natural generalization of classical focusing theorems in what is known as the "Quantum Focusing Conjecture" (QFC). However the quantity that is constrained by the QFC, the generalized entropy, is not an observable due its dependence on an entanglement entropy. Complementary to these approaches, we will focus on *semi-local*, and *state-independent* conditions on the null energy that can provide the right input into a Penrose-type singularity theorem[1].

One such proposal is the Smeared Null Energy Condition (SNEC) [1], which posits that null energy, averaged over portion of an achronal null geodesic, $\mathbf{T}_{++}^{\text{smear}}$, is bounded by, schematically,

$$\langle \mathbf{T}_{++}^{\text{smear}} \rangle \geq -\frac{1}{32\pi G_N (\delta^+)^2}, \tag{2}$$

where $\delta^+$ is the affine length of the smearing. The SNEC implies a semi-classical, Penrose-type singularity theorem applicable in situations with NEC violation [15]. However, the SNEC suffers from two key issues:

- Except in the context of induced gravity [16] in AdS/CFT, a proof of the SNEC has not appeared in the literature.

---

[1]It is interesting to note a middle ground between these two perspectives: a quantum energy bound whose right-hand side is state-dependent but lower-bounded by a fixed observable with sufficiently tame UV behavior [14].

- In the field theory limit $G_N \to 0$, the bound diverges, and so should become sensitive to the ultra-violet (UV) cutoff of the theory.

In this article we address both of this issues.

Firstly, we put the SNEC on more solid footing by proving the field theory limit of the SNEC for free and relevantly (a.k.a. super-renormalizably) perturbed field theories directly on the light-sheet. We find that the bound takes the schematic form

$$\langle \mathbf{T}_{++}^{\text{smear}} \rangle \geq -\frac{\mathcal{N}}{a^{d-2}(\delta^+)^2} , \tag{3}$$

where $\mathcal{N}$ is the number of degrees of freedom, and $\delta^+$ is the smearing length, as before. The above bound, the "field theory SNEC", was proposed by in [1] motivated by the lore that a field theory equipped with a UV cutoff, $a^{-1}$, should have $\frac{\mathcal{N}}{a^{d-2}} \ll \frac{1}{\ell_P^{d-2}}$, where $\ell_P$ is the Planck length (see for instance [17]). To be clear however, this bound is, in principle, a different bound than the original "gravitational SNEC" applying to field theories coupled to semi-classical gravity: the above is a field theoretic bound in Minkowski space with no mention of semi-classical gravity (nor with any mention of $G_N$). Because this bound makes explicit use of a short-distance cutoff, $a$, one should imagine applying it to effective field theories up to an energy scale significantly below $a^{-1}$. Regardless, given the lore stated above, we regard our proof of the "field theory SNEC" as strong credence to the validity of the original, gravitational, SNEC. We will further remark on some utility of this bound in the Discussion, section 4.

Secondly, in the interest of defining an operator that is lower bounded in the continuum limit (that is, without explicit reference to a UV cutoff), we investigate the efficacy of smearing the null energy over more directions. A slightly different phrasing of this inquiry is the following: "Under what conditions can we regard the null energy of an effective field theory as a genuine operator?" For one, we show that no amount of smearing along the transverse light-sheet coordinates provides such a definition. We propose instead that smearing $\mathbf{T}_{++}$ over *two null directions* (a quantity we refer to as the *double-smeared null energy*, or *DSNE*) provides such a definition. We argue that the DSNE is bounded below in free massive theories and propose a general schematic for how this bound scales with the length scales of the smearing. While we motivate this as a "regularization" of the SNEC by an additional smearing over $x^-$, this bound is true for general domains in the $(x^+, x^-)$ plane. We further conjecture that this *double-smeared null energy condition*, or *DSNEC*, remains true for interacting theories and displays a transition as a function of the ratio between the smearing lengths and the correlation length. Schematically, our proposed bound is

$$\langle \mathbf{T}_{++}^{\text{smear}} \rangle \geq -\frac{\mathcal{N} \, C\left(\frac{\delta^+ \delta^-}{\ell^2}\right)}{(\delta^+)^{d/2+1}(\delta^-)^{d/2-1}} , \tag{4}$$

where $\mathcal{N}$ is the number of degrees of freedom, and $\delta^\pm$ are the smearing lengths in the two null directions. $C$ is a function of the dimensionless ratio of the smearing lengths and the correlation length (in the cases we consider it is the inverse of the scalar mass, $\ell^2 = m^{-2}$); we will argue that it is $O(1)$ (with respect to $\mathcal{N}$ and $a$) for nice enough smearing functions. When the mass vanishes it is an $O(1)$ constant. However when $\ell^{-2}\delta^+\delta^- \gg 1$, we will see that $C$ can become damped and provide an even tighter bound. This dovetails nicely with known damping of negative $\mathbf{T}_{\mu\nu}u^\mu u^\nu$ (where $u^\mu$ is a future-directed time-like vector) expectation values for massive scalar theories [18].

Finally, let us pause to mention the following at the onset: while motivated by the role of null energy conditions in semi-classical gravity, the results of this article are purely field theoretical. In particular, Newton's constant will not make an appearance in any of our key equations. We will leave a fuller consideration of the DSNEC in the context of semi-classical gravity to future investigations.

An overview of the organization of this paper follows: we first review necessary facts about quantizing free bosons on a light-sheet and use the "pencil" decomposition of the theory to bootstrap the smeared massless 2d bound to generic dimensions. We then construct an explicit class of squeezed states that realize this bound, at least parametrically with the UV cutoff. After this we propose the lower-boundedness of the DSNEC and argue for its validity through two methods: firstly we calculate its vacuum two-point function and show that it is bounded and secondly we reexamine the DSNE in the same class of smeared states saturating the SNEC. We show that it can also be dimensionally reduced to an expectation value in a 2d *massive* theory and use this prove its lower-boundedness. Along the way we prove a useful family of bounds on the null-energy in 2d massive theories. Finally we end with a discussion of these results, their interplay with interactions, and what further research they suggest.

**A note on conventions**

In this article we will work in $d$ dimensional Minkowski space with the "mostly plus" signature with natural units ($c = \hbar = 1$)

$$ds^2 = -(dx^0)^2 + \sum_{i=1}^{d-1}(dx^i)^2 = -dx^+ dx^- + \sum_{i=2}^{d-1}(dy_\perp^i)^2, \tag{5}$$

where $x^\pm = x^0 \pm x^1$ are lightcone coordinates. As suggested in the above equation, we will typically reserve $\vec{y}_\perp$ for $d-2$ transverse coordinates. Null derivatives are

$$\partial_\pm := \frac{1}{2}(\partial_0 \pm \partial_1), \tag{6}$$

such that $\partial_\pm x^\pm = 1$ and $\partial_\mp x^\pm = 0$. In momentum space we will denote

$$k_\pm = \frac{1}{2}(k_0 \pm k_1), \tag{7}$$

such that $k_+ x^+ + k_- x^- + \vec{k}_\perp \cdot y_\perp = k_\mu x^\mu$. Note that in these conventions the integration measures $dx^+ dx^- = 2dx^0 dx^1$ and $dk_+ dk_- = \frac{1}{2}dk_0 dk_1$; we will denote these by $d^2x^\pm$ and $d^2k_\pm$ for shorthand and to be clear with factors of two. Lastly, we will be contracting the stress tensor along a null-vector, $v_+^\mu$. To be definite, we will denote

$$\mathbf{T}_{++} = v_+^\mu v_+^\nu \mathbf{T}_{\mu\nu}, \qquad v_+^\mu = \frac{1}{2}(1, 1, \vec{0}_\perp). \tag{8}$$

## 2 A lightsheet derivation of the SNEC

We begin with a derivation of the smeared null-energy condition in free scalar field theory through the method of light-sheet quantization [19,20]. The conformal properties of free fields on quantized on a lightsheet make this an powerful approach and similar techniques have been used in proofs of the generalized second law [20], the QNEC in free theories [9,21,22] and

Rényi QNEC variants [23]. We will not repeat the groundwork, which can be found in detail in [20] but state the necessary facts as we go. While we will focus on the story for scalars; we expect a very similar story to apply for spinors and vector bosons following section 4 of [20][2].

The core statement of light-sheet quantization is that free fields are *ultra-local* in tranverse directions when quantized on the lightsheet: the operator algebra and the vacuum state are tensor products of algebras and vacua associated to each null generator. To make this well-defined and explicit, it is useful to discretize the transverse directions of the light-sheet and view each null-generator as a finite width "pencil" of transverse area $a^{d-2}$. This pencil area will play the role of the (inverse) UV cutoff[3]. That is to say we take a lightsheet defined by $\mathcal{L} = \{x^\mu \in \mathbb{R}^{1,d-1} \mid x^- = 0\}$ and realize it as the union of a countable set of these pencils:

$$\mathcal{L} = \cup_{\mathfrak{p}} \mathcal{P}_{\mathfrak{p}}, \tag{9}$$

where $\mathfrak{p}$ labels the pencil $\mathcal{P}_{\mathfrak{p}} = \mathbb{R}_{x^+} \times D_{\mathfrak{p}}$ for small "pixel" $D_{\mathfrak{p}}$ in the transverse directions.

Ultra-locality is then the statement that the operator content is that of a collection of 2d chiral bosonic CFTs, each local to a pencil:

$$\left[ \Phi(x_1^+, \vec{y}_{\mathfrak{p}}), \partial_+ \Phi(x_2^+, \vec{y}_{\mathfrak{p}'}) \right] = \frac{i}{2a^{d-2}} \delta_{\mathfrak{p}\mathfrak{p}'} \delta(x_1^+ - x_2^+), \tag{10}$$

and that the global vacuum decomposes as

$$|\Omega\rangle_{\mathcal{L}} = \bigotimes_{\mathfrak{p}} |\Omega\rangle_{\mathfrak{p}}, \tag{11}$$

where each $|\Omega\rangle_{\mathfrak{p}}$ is the unique null-translation invariant pencil vacuum annihilated by $P_{+,\mathfrak{p}} = \int dx^+ \partial_+ \hat{\Phi}(x^+, \vec{y}_{\mathfrak{p}}) \partial_+ \hat{\Phi}(x^+, \vec{y}_{\mathfrak{p}})$.

A given 2d "pencil CFT" has a stress-tensor $T_{++,\mathfrak{p}}$ which is related to the bulk stress tensor, $\mathbf{T}_{++}$, pulled back to $\mathcal{L}$ by

$$T_{++,\mathfrak{p}}(x^+) = a^{d-2} \mathbf{T}_{++}(x^+, x^- = 0, \vec{y}_{\mathfrak{p}}). \tag{12}$$

Now we can write a generic state on the light-sheet in a way that singles out a particular pencil, $\mathcal{P}_{\bar{\mathfrak{p}}}$:

$$\rho = \sum_{ij} \left( \rho_{\bar{\mathfrak{p}}}^{\Omega} \sigma_{ij} \right) \otimes \rho_{aux}^{\Omega} |i\rangle_{aux} \langle j|_{aux} + \text{h.c.}, \tag{13}$$

where $\rho_{\bar{\mathfrak{p}}}^{\Omega} = |\Omega\rangle_{\bar{\mathfrak{p}}} \langle \Omega|_{\bar{\mathfrak{p}}}$ is the vacuum on the pencil, $\mathcal{P}_{\bar{\mathfrak{p}}}$, and $\sigma_{ij}$ should be thought of as the summation of all possible operator insertions on the pencil $\mathcal{P}_{\bar{\mathfrak{p}}}$: $\sigma_{ij} \sim \delta_{ij} + a^{\frac{d-2}{2}} \int f_{ij} \partial_+ \hat{\Phi} + a^{d-2} \times \int g_{ij} \partial_+ \hat{\Phi} \partial_+ \hat{\Phi} + \dots$. This term also controls the entanglement of the state reduced to $\mathcal{P}_{\bar{\mathfrak{p}}}$ with the other pencils, here labelled as "*aux*." Without loss of generality we parametrize this "*aux*" system by pulling out the tensor product of all the vacuua on its pencils, $\rho_{aux}^{\Omega}$, and take the

---

[2]Namely that much like the free scalar, free spinors and Abelian gauge fields admit an ultra-local pencil decomposition. For the spinor each pencil supports $N_f/2$ 2d massless chiral fermion CFTs (where $N_f$ is the number of components of the $d$ dimensional spinor). For Abelian gauge fields, their pencil theory is of $d-2$ 2d massless decoupled scalars. Importantly, for each theory the bulk null stress tensor is related to corresponding pencil stress tensor by $a^{2-d}$.

[3]To be precise: if we restrict ourselves to a field configurations with transverse momenta $|\vec{p}_\perp| \ll a^{-1}$ then we expand a typical field configuration in a basis of top-hat functions of width $a$. This is explained in appendix A.

basis states $|i\rangle_{aux}$ to be eigenstates of $K_{aux} = -\frac{1}{2\pi} \log \rho_{aux}^{(vac)}$ with eigenvalue $\kappa_i$. The expectation value of the null stress tensor valued on the pencil $\mathcal{P}_{\vec{\mathfrak{p}}}$ can be then written as a sum of 2d CFT stress tensor expectation values:

$$\left\langle \mathbf{T}_{++}(x^+, 0, \vec{y}_{\vec{\mathfrak{p}}}) \right\rangle_\rho = a^{2-d} \sum_i e^{-2\pi\kappa_i} \langle T_{++,\vec{\mathfrak{p}}}(x^+) \rangle_{\rho_{\vec{\mathfrak{p}}}^\Omega \sigma_{ii} + \text{h.c.}} . \tag{14}$$

Now let us define the smeared null-energy (SNE). We will take a square-integrable function on the real line, $f(s)$, normalized with $\int_{-\infty}^{\infty} ds\, f(s)^2 = 1$, and dropping quickly to zero for $s \gg 1$.

$$\mathbf{T}_{++}[f](\vec{y}) := \int \frac{dx^+}{\delta^+} f(x^+/\delta^+)^2 \mathbf{T}_{++}(x^+, x^- = 0, \vec{y}). \tag{15}$$

Here $\delta^+$ controls the length-scale of the smearing[4]. In these conventions, $\mathbf{T}_{++}[f]$ has the same engineering dimensions ($d$) and spin (2) as $\mathbf{T}_{++}$. The SNE expectation value in our light-sheet state is then

$$\langle \mathbf{T}_{++}[f] \rangle_\rho = a^{2-d} \sum_i e^{-2\pi\kappa_i} \int_{-\infty}^{\infty} \frac{dx^+}{\delta^+} f(x^+/\delta^+)^2 \langle T_{++,\vec{\mathfrak{p}}}(x^+) \rangle_{\rho_{\vec{\mathfrak{p}}}^{(vac)} \sigma_{ii} + \text{h.c.}} . \tag{16}$$

Because each $\langle T_{++,\vec{\mathfrak{p}}} \rangle$ is really a 2d CFT expectation value we now leverage the methods of [24] (we will provide an alternative to this method in appendix B). That is along the pencil, we perform the coordinate transformation $x^+ \to u(x^+)$ (with $u$ monotonically increasing with $x^+$). There then exists a unitary operator acting local to the pencil such that

$$\hat{\mathcal{U}}_{\vec{\mathfrak{p}}}^\dagger[u] T_{++,\vec{\mathfrak{p}}}(x^+) \hat{\mathcal{U}}_{\vec{\mathfrak{p}}}[u] = u'(x^+)^2 T_{++,\vec{\mathfrak{p}}}(u(x^+)) - \frac{1}{24\pi} \{u(x^+), x^+\} , \tag{17}$$

with $\{u, x^+\} = \frac{u'''}{u'} - \frac{3}{2}\left(\frac{u''}{u'}\right)^2$ the familiar Schwarzian derivative. Choosing $u'(x^+) = f(x^+/\delta^+)^{-2}$ uniformizes the smearing over the stress tensor and yields the pencil ANEC operator plus an anomaly:

$$\hat{\mathcal{U}}_{\vec{\mathfrak{p}}}^\dagger \mathbf{T}_{++}[f] \hat{\mathcal{U}}_{\vec{\mathfrak{p}}} = a^{2-d} \int_{-\infty}^{\infty} \frac{du}{\delta^+} \hat{\mathbf{T}}_{++,\vec{\mathfrak{p}}}(u, 0, \vec{y}_{\mathfrak{p}}) - \frac{1}{12\pi a^{d-2}} \int_{-\infty}^{\infty} \frac{dx^+}{\delta^+} \left(\frac{d}{dx^+} f(x^+/\delta^+)\right)^2 . \tag{18}$$

The ANEC operator is a positive operator and so the spectrum of $\mathbf{T}_{++}[f]$ is bounded below by the second Schwarzian term. Thus, noting the normalization of $\rho$, $\sum_i e^{-2\pi\kappa_i} \left( \rho_{\vec{\mathfrak{p}}, vac} \sigma_{ii} + \text{h.c.} \right) = 1$, we have

$$\boxed{\langle \mathbf{T}_{++}[f] \rangle_\rho \geq -\frac{1}{12\pi a^{d-2}(\delta^+)^2} \int_{-\infty}^{\infty} ds \left(f'(s)\right)^2 .} \tag{19}$$

This is precisely the SNEC with the pencil area, $a^{d-2}$, replacing $G_N$ as the (inverse) UV cutoff. We pause to note that for decoupled theories[5] where the full stress tensor is the sum of that of each field, then the total number degrees of freedom, $\mathcal{N}$, will multiply the right-hand side of (19). As an example, for $n_s$ scalars, $n_f$ spinors (of $N_f$ components) and $n_v$ Abelian gauge fields we will have

$$\mathcal{N} = n_b + n_s \frac{N_f}{2} + (d-2)n_v . \tag{20}$$

See footnote 2 for elaboration on this counting.

---

[4]To precisely fix this scaling in definite terms we will further fix the standard deviation of $f^2$ (thought of as a probability distribution over $\mathbb{R}$) to $\sigma_f := \int_{-\infty}^{\infty} ds\, s^2 f(s)^2 - \left(\int_{-\infty}^{\infty} ds\, s f(s)^2\right)^2 = 1$

[5]In fact, the fields can also be coupled as long as the coupling is relevant. See the discussion on interactions in section 4.

## 2.1 The futility of transverse smearing

One might hope that the current SNEC, (19), can be strengthened by appropriately smearing in the transverse direction:

$$\mathbf{T}_{++}[\mathcal{F}] = \int_{-\infty}^{\infty} \frac{dx^+}{\delta^+} \int \frac{d^{d-2}y_\perp}{\mathcal{A}^{d-2}} \, \mathcal{F}^2\left(\frac{x^+}{\delta^+}, \frac{\vec{y}_\perp}{\mathcal{A}}\right) \mathbf{T}_{++}(x^+, x^- = 0, \vec{y}_\perp), \qquad (21)$$

for some smooth square integrable function $\mathcal{F}(s^+, \vec{s}_\perp)$ on $\mathcal{L}$ that falls off quickly as $|s^+|, |\vec{s}_\perp| \gg 1$ and normalized[6] to $\int ds^+ \int d^{d-2}\vec{s}_\perp \mathcal{F}(s^+, \vec{s}_\perp)^2 = 1$. It is useful to keep in mind as a particular example the Gaussian smearing function,

$$\mathcal{F}^{\text{Gauss}}(s^+, \vec{s}_\perp) = \frac{1}{(2\pi)^{\frac{d-1}{4}}} e^{-\frac{s_+^2}{4}} e^{-\frac{|\vec{s}_\perp|^2}{4}}. \qquad (22)$$

Indeed since the smearing over $d - 2$ transverse directions introduces another length scale, $\mathcal{A}^{d-2}$, it seems possible, at least on dimensional grounds, that this could lead to a bound independent of the UV cutoff. We now show that this is not the case. In fact we find that (19) carries over naturally:

$$\langle \mathbf{T}_{++}[\mathcal{F}] \rangle \geq -\frac{1}{12\pi \, a^{d-2}(\delta^+)^2} \int d^{d-2}\vec{s}_\perp \int ds^+ \left(\partial_{s^+} \mathcal{F}(s^+, \vec{s}_\perp)\right)^2. \qquad (23)$$

We will proceed firstly with a general argument from light-sheet quantization and then we will construct example states that defy strengthening the SNEC through transverse smearing.

Proceeding forward, let us discretize our transverse smearing as a summation over pencils, again of area $a^{d-2}$, the inverse of the UV cutoff:

$$\langle \mathbf{T}_{++}[\mathcal{F}] \rangle_\rho \simeq \frac{1}{N} \sum_{\mathfrak{p}} \int \frac{dx^+}{\delta^+} \mathcal{F}_{(\mathfrak{p})}^2(x^+/\delta^+) \langle \mathbf{T}_{++}(x^+, 0, \vec{y}_\mathfrak{p}) \rangle_\rho. \qquad (24)$$

For the sake of this argument we will take $\mathcal{F}$ to have compact support in the transverse directions and $\mathcal{A}^{d-2}/a^{d-2} = N$ is the number of pencils that $\mathcal{F}$ has support on. Focusing now on this collection of pencils $\{\mathcal{P}_\mathfrak{p}\}_{\mathfrak{p}=1,\dots,N}$, let us reparameterize our state $\rho$ as:

$$\rho = \sum_{ij} \left(\rho_1^\Omega \otimes \dots \otimes \rho_N^\Omega\right) \Sigma_{ij} \otimes \rho_{aux}^\Omega |i\rangle_{aux} \langle j|_{aux} + h.c.. \qquad (25)$$

The operator $\Sigma_{ij}$ has the dual purpose of entangling the collection of pencils $\{\mathcal{P}_\mathfrak{p}\}$ with the "$aux$" pencils, as well as summing up all of the operator insertions on $\{\mathcal{P}_\mathfrak{p}\}$ preparing the state. For the derivation of the bound in section 2, we were agnostic about the details of these operator insertions. However, as was handily noted in [9], $\Sigma_{ij}$ admits an expansion about the vacuum with $n$-particle contributions having coefficients that scaling as $a^{n\frac{d-2}{2}}$. Thus for small $a$[7] the leading contribution to the stress-tensor expectation value comes from two particle insertions. Furthermore, it is easy to see that $\langle \mathbf{T}_{++}(x^+, 0, \vec{y}_\mathfrak{p}) \rangle_\rho$ can only be non-zero if $both$ of those insertions occur on the same pencil, $\mathcal{P}_\mathfrak{p}$. Thus the only relevant contributions to $\Sigma_{ij}$ are of the form

$$\Sigma_{ij} \supset \delta_{ij} + \sum_{\mathfrak{p}=1}^{N} \sigma_{ij}^{(\mathfrak{p})} + \dots, \qquad (26)$$

---

[6] As before, we will also normalize the standard deviations of $\mathcal{F}$ as $\int ds^+ \int d^{d-2}s_\perp (s^+)^2 \mathcal{F}^2 - \left(\int ds^+ \int d^{d-2}s_\perp s^+ \mathcal{F}^2\right)^2 = \int ds^+ \int d^{d-2}s_\perp (s_\perp)^2 \mathcal{F}^2 = 1$.

[7] ...compared to the characteristic wavelengths of the state.

where $\sigma_{ij}^{(\mathrm{p})}$ is a collection of operator insertions on $\mathcal{P}_\mathfrak{p}$ beginning with 2-particle insertions. Thus we see that in the small $a$ limit the relevant contributions to $\langle \mathbf{T}_{++} \rangle$ are diagonal in the pencils

$$\langle \mathbf{T}_{++}(x^+, 0, \vec{y}_\mathfrak{p}) \rangle_\rho = \sum_i e^{-2\pi\kappa_i} \langle \mathbf{T}_{++}(x^+, 0, \vec{y}_\mathfrak{p}) \rangle_{\rho_\mathfrak{p}^\Omega \sigma_{ii}^{(\mathrm{p})} + \mathrm{h.c.}} + \dots. \tag{27}$$

We now know that smearing each one of these expectation values is bounded by the right-hand side of (19). Thus we arrive at

$$\frac{1}{N} \sum_\mathfrak{p} \int \frac{dx^+}{\delta^+} \mathcal{F}_{(\mathrm{p})}^2(x^+/\delta^+) \langle \mathbf{T}_{++}(x^+, 0, \vec{y}_\mathfrak{p}) \rangle_\rho \geq -\frac{1}{12\pi N a^{d-2}(\delta^+)^2} \sum_\mathfrak{p} \int_{-\infty}^\infty ds^+ \left( \mathcal{F}_{(\mathrm{p})}'(s^+) \right)^2. \tag{28}$$

Rewriting this sum as $\sum_\mathfrak{p} \sim \frac{1}{a^{d-2}} \int d^{d-2}\vec{y}$ we have

$$\langle \mathbf{T}_{++}[\mathcal{F}] \rangle_\rho \geq -\frac{1}{12\pi a^{d-2}(\delta^+)^2} \int d^{d-2}\vec{s}_\perp \int ds^+ \left( \partial_{s^+} \mathcal{F}(s^+, \vec{s}_\perp) \right)^2. \tag{29}$$

Note that if $\mathcal{F}$ factorizes $\mathcal{F}(s^+, \vec{s}_\perp) = \mathcal{F}_+(s^+)\mathcal{F}_\perp(\vec{s}_\perp)$, e.g. $\mathcal{F}^{\mathrm{Gauss}}$, then the transverse integral drops out explicitly.

## 2.2 A series of squeezed states

Let us illustrate that (29) is not simply a failure of the pencil construction to strengthening the SNEC by constructing a set of squeezed states with tunable negative null energy up to the UV cutoff. The role of squeezed states (originally studied in the context of quantum optics [25]) in realizing negative energy densities in quantum field theory is well known [26–28]. To construct the states, we first note the null Fock quantization of the fields

$$\Phi(x^+, 0, \vec{y}) = \int_0^\infty \frac{dk_+}{(2\pi)\sqrt{2k_+}} \int \frac{d^{d-2}\vec{p}_\perp}{(2\pi)^{d-2}} \left( \hat{a}_{k_+, \vec{p}_\perp} e^{-ik_+ x^+ - i\vec{p}_\perp \cdot \vec{y}} + \hat{a}_{k_+, \vec{p}_\perp}^\dagger e^{ik_+ x^+ + i\vec{p}_\perp \cdot \vec{y}} \right), \tag{30}$$

with commutators

$$[\hat{a}_{k_+, \vec{p}_\perp}, \hat{a}_{k_+', \vec{p}_\perp'}^\dagger] = (2\pi)\delta(k_+ - k_+')(2\pi)^{d-2}\delta^{d-2}(\vec{p}_\perp - \vec{p}_\perp'), \tag{31}$$

and mass-shell condition $k_- = \frac{\vec{p}_\perp^2 + m^2}{4k_+}$ and $k_+ \geq 0$. The normal ordering in $\mathbf{T}_{++}$ is with respect to this Fock quantization.

The squeezed state in question is defined by a $\mathbb{C}$-valued symmetric bi-function of momenta $\xi(k_+^1, \vec{p}_\perp^1; k_+^2, \vec{p}_\perp^2) = \xi(k_+^2, \vec{p}_\perp^2; k_+^1, \vec{p}_\perp^1)$:

$$|\xi\rangle := \hat{S}[\xi]|\Omega\rangle_\mathcal{L}, \tag{32}$$

where

$$\hat{S}[\xi] = \exp\left( \frac{1}{2} \left( \hat{a}^\dagger \circ \xi \circ \hat{a}^\dagger - \hat{a} \circ \xi^* \circ \hat{a} \right) \right), \tag{33}$$

and we have introduced a distributional "matrix" notation, "$\circ$," on momentum space bi-functions

$$\left( \tilde{f}_1 \circ \tilde{f}_2 \right)(k_+^1, \vec{p}_\perp^1; k_+^2, \vec{p}_\perp^2) = \int \frac{dk_+}{2\pi} \int \frac{d^{d-2}\vec{p}_\perp}{(2\pi)^{d-2}} \tilde{f}_1(k_+^1, \vec{p}_\perp^1; k_+, \vec{p}_\perp) \tilde{f}_2(k_+, \vec{p}_\perp; k_+^2, \vec{p}_\perp^2). \tag{34}$$

In what follows we will take $\xi \in \mathbb{R}$ for notational simplicity. Since $\hat{S}[\xi]$ is unitary, our state $|\xi\rangle$ is normalized. It is a simple enough exercise to show that $\hat{S}[\xi]$ acts by conjugation on the Fock modes as

$$\hat{S}[\xi]^{\dagger}\hat{a}_{k_+,\vec{p}_\perp}\hat{S}[\xi] = \left(\cosh_\circ[\xi] \circ \hat{a}\right)_{k_+,\vec{p}_\perp} + \left(\sinh_\circ[\xi] \circ \hat{a}^{\dagger}\right)_{k_+,\vec{p}_\perp}, \tag{35}$$

and on $\hat{a}^{\dagger}$ by the Hermitian conjugation of the above. $\cosh_\circ[\xi]$ and $\sinh_\circ[\xi]$ are defined by their Taylor expansion with the $\circ$ product defined in (34). Using this it is easy to evaluate $\mathbf{T}_{++}[\mathcal{F}]$ in this state:

$$\langle\xi|\mathbf{T}_{++}[\mathcal{F}]|\xi\rangle = \int \frac{d^{d-2}\vec{p}_\perp d^{d-2}\vec{p}'_\perp}{(2\pi)^{2d-4}} \int \frac{dk_+ dk'_+}{(2\pi)^2}(k_+ k'_+)^{\frac{1}{2}}\left(\sinh_\circ^2(\xi)(k_+,\vec{p}_\perp;k'_+,\vec{p}'_\perp)\mathfrak{R}\tilde{\mathcal{G}}_{\delta^+\Delta_+,\mathcal{A}\Delta_\perp}\right.$$
$$\left. - \sinh_\circ(\xi)\circ\cosh_\circ(\xi)(k_+,\vec{p}_\perp;k'_+,\vec{p}'_\perp)\mathfrak{R}\tilde{\mathcal{G}}_{\delta^+\Sigma_+,\mathcal{A}\Sigma_\perp}\right), \tag{36}$$

where $\mathfrak{R}$ stands for the real part, $\tilde{\mathcal{G}}$ is the Fourier transform of $\mathcal{F}^2$ (in dimensionless variables):

$$\tilde{\mathcal{G}}_{\rho_+,\vec{\rho}_\perp} = \int ds^+ \int d^{d-2}\vec{s}_\perp \,\mathcal{F}(s^+,\vec{s}_\perp)^2 e^{-is^+\rho_+ - i\vec{s}_\perp\cdot\vec{\rho}_\perp}, \tag{37}$$

and

$$\Delta_+ = k_+ - k'_+, \qquad \Delta_\perp = \vec{p}_\perp - \vec{p}'_\perp, \qquad \Sigma_+ = k_+ + k'_+, \qquad \Sigma_\perp = \vec{p}_\perp + \vec{p}'_\perp. \tag{38}$$

We are interested in how low we can tune $\langle\mathbf{T}_{++}[\mathcal{F}]\rangle$ by tuning $\xi$. Although partially hidden by our notation, this is a complex minimization problem. We will simplify things by positing an ansatz for $\xi$ motivated by the following physical reasoning: the negative null energy can apparently become arbitrarily negative when the damping in $\Sigma_\perp$ fails. This is precisely when the state is composed of particles of *large and oppositely oriented transverse momenta* (the absence of this transverse momenta is why the smeared negative null energy in two dimensions remains $O(1)$). Thus we will look at states with

$$\xi(k_+,\vec{p}_\perp,k'_+,\vec{p}'_\perp) = \xi(k_+,k'_+,|\vec{p}_\perp|)(2\pi)^{d-2}\delta^{d-2}(\vec{p}_\perp + \vec{p}'_\perp). \tag{39}$$

Within this ansatz, the $n^{th}$ power (with respect to the $\circ$ product) of $\xi$ is:

$$\left(\xi\right)_\circ^n(k_+,\vec{p}_\perp,k'_+,\vec{p}'_\perp) = (\xi)_\bullet^n(k_+,k'_+;|\vec{p}_\perp|)(2\pi)^{d-2}\delta^{d-2}(\vec{p}_\perp - (-1)^n\vec{p}'_\perp), \tag{40}$$

where we've introduced another matrix notation, "$\bullet$," for $k_+$ integrations:

$$(\tilde{f}_1 \bullet \tilde{f}_2)(k_+^1,k_+^2) := \int \frac{dk_+}{2\pi}\tilde{f}_1(k_+^1;k_+)\tilde{f}_2(k_+;k_+^2). \tag{41}$$

Due to the difference in the even and odd powers of $\xi$ this ansatz has the effect of completely nullifying the dependence of the smearing functions on the transverse momenta:

$$\langle\mathbf{T}_{++}[\mathcal{F}]\rangle_\xi = \int \frac{d^{d-2}\vec{p}_\perp}{(2\pi)^{d-2}} \int_0^\infty \frac{dk_+ dk'_+}{(2\pi)^2}(k_+ k'_+)^{1/2}\left(\sinh_\bullet^2(\xi)(k_+,k'_+;|\vec{p}_\perp|)\mathfrak{R}\tilde{\mathcal{G}}_{\delta^+\Delta_+,0}\right.$$
$$\left. - \sinh_\bullet(\xi)\bullet\cosh_\bullet(\xi)(k_+,k'_+;|\vec{p}_\perp|)\mathfrak{R}\tilde{\mathcal{G}}_{\delta^+\Sigma_+,0}\right). \tag{42}$$

As such, one might suspect it is possible to relate (42) to an expectation value in an appropriate 2d theory defined along a light-ray. This is indeed the case as we show now. Consider the smeared null energy of the scalar in two dimensions, restricted to the light-sheet, $\mathcal{L}$, at $x^- = 0$:

$$T_{++}[f] = \int \frac{dx^+}{\delta^+}f\left(\frac{x^+}{\delta^+}\right)^2 :\partial_+\varphi\partial_+\varphi:(x^+),$$
$$\varphi(x^+,0) = \int \frac{dk_+}{2\pi\sqrt{2k_+}}\left(\hat{a}_{k_+}e^{-ik_+x^+} + \hat{a}_{k_+}^{\dagger}e^{ik_+x^+}\right), \tag{43}$$

and a 2d squeezed state[8]

$$|\xi(\mu)\rangle_{2d} = \hat{s}[\xi]|\Omega\rangle_{2d}, \qquad \hat{s}[\xi] = \exp\left(\frac{1}{2}\int \frac{dk_+^1 dk_+^2}{(2\pi)^2} \xi(k_+^1, k_+^2, \mu)\hat{a}_{k_+^1}^\dagger \hat{a}_{k_+^2}^\dagger - \text{h.c.}\right). \qquad (44)$$

Here $\mu$ appears as an auxiliary parameter in the squeezing function that we will be interested in tuning within an ensemble of squeezed states. By similar manipulations to above, it is then easy to show that indeed

$$\langle \mathbf{T}_{++}[\mathcal{F}]\rangle_\xi = \frac{V_{d-3}}{(2\pi)^{d-3}} \int_0^\infty \frac{d|\vec{p}_\perp|}{2\pi} |\vec{p}_\perp|^{d-3} \langle \xi(\mu)|T_{++}[f]|\xi(\mu)\rangle_{2d}|_{\mu=|\vec{p}_\perp|}, \qquad (45)$$

with a 2d smearing function related to $d$-dimensional smearing function via

$$f(s^+)^2 = \int d^{d-2}\vec{s}_\perp \mathcal{F}(s^+, \vec{s}_\perp)^2, \qquad (46)$$

and $V_{d-3} = \frac{2\pi^{\frac{d-2}{2}}}{\Gamma(\frac{d-2}{2})}$ is the surface volume of a $d-3$ sphere. Because we are free to choose the functional dependence of the squeezing function $\xi$ on $|\vec{p}_\perp|$, it should be clear that (45) has the very real danger of diverging due to this transverse momenta. For instance, to make this explicit we could choose

$$\xi(k_+, k_+', |\vec{p}_\perp|) = \chi(k_+, k_+')\Theta_{M,\Delta M}(|\vec{p}_\perp|), \qquad (47)$$

where $\Theta_{M,\Delta_M}(|\vec{p}_\perp|)$ is a Heaviside function with support on a shell of transverse momenta centered at $M$ and of width $\Delta M$. Because $\Theta$ squares to itself (at least distributionally), $(\xi)_\bullet^n = (\chi)_\bullet^n \times \Theta_{M,\Delta M}$ and so

$$\langle \mathbf{T}_{++}[\mathcal{F}]\rangle_\xi = \frac{V_{M,\Delta M}}{(2\pi)^{d-2}} \langle \chi|T_{++}[f]|\chi\rangle_{2d}, \qquad (48)$$

where $V_{M,\Delta M} = \int d^{d-2}\vec{p}_\perp \, \Theta_{M,\Delta M}$ is the volume of the transverse momentum space shell. For thin shells, $\Delta M/M \ll 1$ it scales like $M^{d-2}$:

$$V_{M,\Delta M} \approx \frac{2\pi^{\frac{d-2}{2}}}{\Gamma(\frac{d-2}{2})}\left(\frac{\Delta M}{M}\right)M^{d-2}. \qquad (49)$$

This is regardless of the details on how we choose the $\chi(k_+, k_+')$ squeezing parameter. Because $|\chi\rangle_{2d}$ itself is a squeezed state in 2d, it is easy to arrange that

$$\langle \chi|T_{++}[f]|\chi\rangle_{2d} = \int_0^\infty \frac{dk_+ dk_+'}{(2\pi)^2} (k_+ k_+')^{1/2}\left(\sinh_\bullet^2(\chi)(k_+, k_+')\Re\widetilde{(f^2)}_{\delta^+ \Delta_+}\right.$$
$$\left. - \sinh_\bullet(\chi)\bullet\cosh_\bullet(\chi)(k_+, k_+')\Re\widetilde{(f^2)}_{\delta^+ \Sigma_+}\right) < 0, \qquad (50)$$

say by making the overall magnitude of $\chi$ small so that the second term dominates.[9] In this case the full $d$-dimensional expectation value will also be negative, with the additional $V_{M,\Delta M}$

---

[8]We are still taking $\xi \in \mathbb{R}$, which precludes some generality. We are also ignoring the possible inclusion of a "displacement operator" $D[\theta] = \exp\left(\int \frac{d\kappa}{2\pi}\theta_\kappa \hat{a}_\kappa - \text{h.c.}\right)$. Theses exclusions will not affect our conclusions.

[9]In fact for small $\chi$, one can imagine expanding the exponential in (44) to first order in $\chi$. The UV divergent negative null energy is then related to a divergent negative null-energy in "0+2" particle states noted in [29].

coming for the ride. Thus it seems possible to engineer states with $\langle \mathbf{T}_{++}[\mathcal{F}]\rangle \sim -M^{d-2}$. Because $\langle T_{++}[f]\rangle_{2d}$ is bounded below by $-\frac{1}{12\pi(\delta^+)^2}\int ds^+ f'(s^+)^2$ for all states we can transplant this to a bound on $\langle \mathbf{T}_{++}[\mathcal{F}]\rangle_\xi$, at least for this series of squeezed states

$$\langle \mathbf{T}_{++}[\mathcal{F}]\rangle_\xi \geq -c(\Delta M/M)\frac{M^{d-2}}{(\delta^+)^2}\int_{-\infty}^{\infty} ds^+ \frac{\left(\int d^{d-2}\vec{s}_\perp \mathcal{F}(s^+,\vec{s}_\perp)\partial_{s^+}\mathcal{F}\left(s^+,\vec{s}_\perp\right)\right)^2}{\int d^{d-2}\vec{s}_\perp \mathcal{F}\left(s^+,\vec{s}_\perp\right)^2}, \qquad (51)$$

where $c(\Delta M/M) = \frac{V_{\Delta M,M}}{6(2\pi)^{d-1}M^{d-2}} \approx \frac{2\Delta M/M}{3(4\pi)^{\frac{d}{2}}\Gamma\left(\frac{d-2}{2}\right)}$. We can consider a series of states with increasing $M$ and with small but *fixed* $\Delta M/M$. The only limiting factor on this series of states is the validity of effective field theory, which is to say that we don't excite states with momentum on the order of $a^{-1}$. Because of this, we find that is not possible to improve the bound (29)[10], by an order of the UV cutoff.

## 3 Double null smearing

So far we have seen that even for free scalar theories the null stress-tensor restricted to light-sheet, $\mathcal{L}$, fails to be a lower-bounded operator when the UV cutoff, $a^{-1}$, is taken to infinity. It is perhaps clear that if we want a lower bound on null-energy that is *both* (i) state independent and (ii) independent of the UV cutoff then we will have to smear off the lightsheet. In this section, following this logic, we investigate the null-energy smeared along both null-rays in what we will coin the *DSNE* (for "double smeared null-energy"). Indeed, smearing in both $x^+$ and $x^-$ is morally similar to smearing in a time-like direction and there is good evidence that the null-energy is well behaved when averaged over a finite time-scale [30]. Firstly, we will investigate the vacuum two-point function of the DSNE of the free massive boson and show that it is bounded by a cutoff independent quantity. While this is does not constitute a proof of the lower-boundedness of the DSNE, it establishes the plausibility of it as a bounded operator. Secondly, by revisiting the same series of squeezed states from section 2.2, we will propose a family of bounds that we will coin the *DSNEC*.

### 3.1 Alternative quantization

Before jumping head-first, let us briefly reorganize the Fock space of modes: (30) and (31) are currently well suited for manipulations of null and transverse momenta, $k_+$ and $\vec{p}_\perp$ (respectively), however we will find it helpful to work explicitly with the set of two null momenta $k_+$ and $k_-$. In the trade-off we loose the norm of the transverse momenta, $|\vec{p}_\perp|$, as a quantum number but retain its direction, $\hat{n}_{\vec{p}_\perp}$ which we conveniently label as a collective set of angles, $\Omega_{d-3}$, on the unit $S^{d-3}$. It is easy to work out that the corresponding quantization is

$$\Phi(x^+,x^-,\vec{y}) = \int_{\mathcal{D}_{m^2}} \frac{d^2k_\pm}{(2\pi)^2} \int \frac{d\Omega_{d-3}}{(2\pi)^{d-3}}(4k_+k_--m^2)^{\frac{d-4}{4}}$$
$$\times \left(\hat{a}_{k_+,k_-,\Omega}e^{-ik_+x^+-ik_-x^--i(k_+k_--m^2)^{1/2}|y|\cos\Omega} + \text{h.c.}\right). \qquad (53)$$

---

[10]The right-hand side of (51) is consistent with (29) via the Cauchy-Schwarz integral inequality:

$$\frac{\left(\int d^{d-2}\vec{s}_\perp \mathcal{F}\partial_{s^+}\mathcal{F}\right)^2}{\int d^{d-2}\vec{s}_\perp \mathcal{F}^2} \leq \int d^{d-2}\vec{s}_\perp \left(\partial_{s^+}\mathcal{F}\right)^2, \qquad (52)$$

with equality when $\mathcal{F}$ factorizes $\mathcal{F}(s^+,\vec{s}_\perp) = \mathcal{F}_+(s^+)\mathcal{F}_\perp(\vec{s}_\perp)$. Regardless, this discrepancy is O(1) and so does not change our conclusion that these squeezed states are concrete counterexamples to strengthening the SNEC by O($\mathcal{A}^{d-2}/a^{d-2}$) by transverse smearing.

The null-momenta are restricted to the domain $\mathcal{D}_{m^2} = \{k_\pm \geq 0 \,\big|\, 4k_+k_- \geq m^2\}$. The oscillators satisfy

$$[\hat{\mathrm{a}}_{k_+,k_-,\Omega}, \hat{\mathrm{a}}^\dagger_{k'_+,k'_-,\Omega'}] = (2\pi)^{d-1}\delta(k_+ - k'_+)\delta(k_- - k'_-)\delta^{d-3}_{S^{d-3}}(\Omega - \Omega'), \tag{54}$$

where $\delta^{d-3}_{S^{d-3}}(\Omega)$ is the normalized delta function on the unit $S^{d-3}$ with the north-pole set as the origin. This Fock set of modes is related to those in (31) as

$$\hat{\mathrm{a}}_{k_+,k_-,\Omega} = \sqrt{2}\, k_+^{1/2}(4k_+k_- - m^2)^{\frac{d-4}{4}}\hat{a}_{k_+,\vec{p}}, \qquad 4k_+k_- - \vec{p}_\perp^2 - m^2 = 0. \tag{55}$$

When it does not cause confusion we will often continue to write $\vec{p}_\perp = (4k_+k_- - m^2)^{1/2}\hat{n}_\Omega$ for notational simplicity, with the tacit understanding that $k_+$, $k_-$, and $\Omega$ are the actual quantum numbers. Now we smear the null stress tensor with a smooth, $L^2$-normalized function $\mathcal{F}(s^+, s^-)$ dropping off quickly[11] for $|s^\pm| \gg 1$:

$$\begin{aligned}
\mathbf{T}_{++}[\mathcal{F}](\vec{y}) &= \int_{-\infty}^\infty \frac{d^2x^\pm}{\delta^+\delta^-}\mathcal{F}(x^+/\delta^+, x^-/\delta^-)^2\mathbf{T}_{++}(x^+, x^-, \vec{y}) \\
&= \int_{\mathcal{D}_{m^2}} \frac{d^2k_\pm d^2k'_\pm}{(2\pi)^4}\int \frac{d\Omega_{d-3}d\Omega'_{d-3}}{(2\pi)^{2d-6}}|\vec{p}_\perp|^{\frac{d-4}{2}}|\vec{p}'_\perp|^{\frac{d-4}{2}}(k_+k'_+) \\
&\quad \times \left\{2\Re\left(\tilde{\mathcal{G}}_{\delta^+\Delta_+,\delta^-\Delta_-}e^{i(\vec{p}-\vec{p}')_\perp\cdot\vec{y}}\right)\hat{\mathrm{a}}^\dagger_{k_+,k_-,\Omega}\hat{\mathrm{a}}_{k'_+,k'_-,\Omega'}\right. \\
&\quad \left.-\left(\tilde{\mathcal{G}}_{\delta^+\Sigma_+,\delta^-\Sigma_-}\hat{\mathrm{a}}_{k_+,k_-,\Omega}\hat{\mathrm{a}}_{k'_+,k'_-,\Omega'}e^{i(\vec{p}+\vec{p}')_\perp\cdot\vec{y}} + \text{h.c.}\right)\right\},
\end{aligned} \tag{56}$$

where

$$\Delta_+ = k_+ - k'_+, \qquad \Delta_- = k_- - k'_-, \qquad \Sigma_+ = k_+ + k'_+, \qquad \Sigma_- = k_- + k'_-, \tag{57}$$

and

$$\tilde{\mathcal{G}}_{\rho_+,\rho_-} = \int_{-\infty}^\infty ds^+ds^-\,\mathcal{F}(s^+, s^-)^2 e^{is^+\rho_+ + is^-\rho_-}. \tag{58}$$

Our first interest is gauging how negative expectation values of $\mathbf{T}_{++}[\mathcal{F}]$ can become. We will proceed by evaluating the two-point function of $\mathbf{T}_{++}[\mathcal{F}]$ in the vacuum. This will give us a rough order of magnitude of how large the fluctuations (both positive and negative) of $\mathbf{T}_{++}[\mathcal{F}]$ can become.

## 3.2 The vacuum two-point function

From (56) we write the vacuum two-point function[12]:

$$\langle(\mathbf{T}_{++}[\mathcal{F}])^2\rangle_\Omega = \frac{2V_{d-3}^2}{(2\pi)^{2(d-3)}}\int_{\mathcal{D}_{m^2}} \frac{d^2k_\pm d^2k'_\pm}{(2\pi)^4}(4k_+k_- - m^2)^{\frac{d-4}{2}}(4k'_+k'_- - m^2)^{\frac{d-4}{2}}k_+^2 k'^2_+\left|\tilde{\mathcal{G}}_{\delta^+\Sigma_+,\delta^-\Sigma_-}\right|^2. \tag{59}$$

There are two regimes that we are interested in: (i) the smearing is much less than the correlation length, $\ell^2 \sim m^{-2}$, ($\delta^+\delta^- m^2 \ll 1$) and (ii) much larger than the correlation length ($\delta^+\delta^- m^2 \gg 1$).

---

[11]Again, we will fix $\int d^2s\,(s^+)^2\mathcal{F}^2 - \left(\int d^2s\,s^+\mathcal{F}^2\right)^2 = \int d^2s\,(s^-)^2\mathcal{F}^2 - \left(\int d^2s\,s^-\mathcal{F}^2\right)^2 = 1$ for definiteness and to fix the smearing lengths $\delta^\pm$.

[12]Evaluated at the same transverse point, $\vec{y}_\perp$, which disappears as a consequence of translation invariance of the vacuum.

Let us first investigate $m^2 = 0$. We rescale $\kappa_\pm = \delta^\pm k_\pm$:

$$
\langle (\mathbf{T}_{++}[\mathcal{F}])^2 \rangle_\Omega \big|_{m^2=0} = \frac{2^{2d-7} V_{d-3}^2}{(2\pi)^{2(d-1)}} (\delta^+)^{-(d+2)} (\delta^-)^{-(d-2)}
$$
$$
\times \int_0^\infty d^2\kappa_\pm d^2\kappa'_\pm (\kappa_-\kappa'_-)^{\frac{d-4}{2}} (\kappa_+\kappa'_+)^{\frac{d}{2}} \left| \tilde{\mathcal{G}}_{\kappa_+ + \kappa'_+, \kappa_- + \kappa_-} \right|^2 . \tag{60}
$$

The integrals over $(\kappa - \kappa')_\pm$ can be done leading to

$$
\langle (\mathbf{T}_{++}[\mathcal{F}])^2 \rangle_\Omega \big|_{m^2=0} = c_d (\delta^+)^{-(d+2)} (\delta^-)^{-(d-2)} \int_0^\infty d^2\rho_\pm, \rho_+^{d+1} \rho_-^{d-3} \left| \tilde{\mathcal{G}}_{\rho_+, \rho_-} \right|^2 , \tag{61}
$$

with $c_d = \frac{1}{8(4\pi)^{d-1}} \frac{\Gamma(\frac{d+2}{2})}{\Gamma(\frac{d-2}{2})\Gamma(\frac{d-1}{2})\Gamma(\frac{d+3}{2})}$. For $|\tilde{\mathcal{G}}|^2$ falling off faster than an appropriate polynomial at large momenta,

$$
\lim_{\rho_+ \to \infty} |\tilde{\mathcal{G}}_{\rho_+, \rho_-}| \lesssim \rho_+^{-\frac{d+2}{2}} ,
$$
$$
\lim_{\rho_- \to \infty} |\tilde{\mathcal{G}}_{\rho_+, \rho_-}| \lesssim \rho_-^{-\frac{d-2}{2}} , \tag{62}
$$

this expression already indicates that $\langle (\mathbf{T}_{++}[\mathcal{F}])^2 \rangle_\Omega$ is finite and that it scales with smearing lengths as $(\delta^+)^{-(d+2)} (\delta^-)^{-(d-2)}$. The subsequent integral is an $O(1)$ contribution controlled by the moments of the smearing function in momentum space. While already a useful result, we might want to investigate (61) in the cases where it can be expressed locally in position space. This is, in general, not possible because the momentum integrals do not extend over the entire plane. However when $d$ is odd and $\mathcal{F}$ factorizes, $\mathcal{F}(s) = \mathcal{F}_+(s^+)\mathcal{F}_-(s^-)$ we can perform the dual Fourier transform to find

$$
\langle (\mathbf{T}_{++}[\mathcal{F}])^2 \rangle_\Omega \big|_{m^2=0} = \pi^2 c_d (\delta^+)^{-(d+2)} (\delta^-)^{-(d-2)} \int ds^+ \mathcal{F}_+^2(s^+) (i\partial_{s^+})^{d+1} \mathcal{F}_+^2(s^+)
$$
$$
\times \int ds^- \mathcal{F}_-^2(s^-) (i\partial_{s^-})^{d-3} \mathcal{F}_-^2(s^-) . \tag{63}
$$

For small $m^2\delta^+\delta^-$ it is certainly possible proceed from (59) perturbatively in $\gamma^2 = m^2\delta^+\delta^-$ through the change of variables, $v = \delta^+\delta^-(4k_+k_- - m^2)$ and $\kappa_+ = \delta^+k_+$ which removes the dependence on $m^2$ in the integration region

$$
\langle (\mathbf{T}_{++}[\mathcal{F}])^2 \rangle_\Omega = \frac{V_{d-3}^2}{8(2\pi)^{2(d-1)}} (\delta^+)^{-(d+2)} (\delta^-)^{-(d-2)}
$$
$$
\times \int_0^\infty d\kappa_+ d\kappa'_+ \int_0^\infty dv\,dv' (vv')^{\frac{d-4}{2}} \kappa_+ \kappa'_+ \left| \tilde{\mathcal{G}}_{\kappa_+ + \kappa'_+, \frac{v+\gamma^2}{4\kappa_+} + \frac{v'+\gamma^2}{4\kappa'_+}} \right|^2 . \tag{64}
$$

It is clear from (64) that expanding $|\tilde{\mathcal{G}}|^2$ in orders of $\gamma^2$ leads to an expansion in $1/\kappa_+ + 1/\kappa'_+$ and so leads to tamer UV behavior. This expansion is somewhat subtle however: for high enough orders the inverse powers of $\kappa_+$ can lead to spurious IR divergences. We view this as an artifact of the perturbation theory: in fact by examining the $\gamma \gg 1$ regime directly, we can see that (64) is both UV and IR convergent for large masses. We do this now.

For small correlation lengths, $\gamma^2 = \delta^+\delta^-m^2 \gg 1$, our primary tool will be to simplify the

integral by saddle-point. The broad strategy is the following: since $|\tilde{\mathcal{G}}(\rho_+, \rho_-)|^2$ is a positive function we will write it as $|\tilde{\mathcal{G}}(\rho_+, \rho_-)|^2 = \exp(-\mathcal{S}(\rho_+, \rho_-))$ for some real $\mathcal{S}$ that grows faster than a logarithm at large argument. It will be useful to introduce another change of variables, $\kappa_+ = \gamma \ell_+$ and $\nu = \gamma^2 w$, to make the saddle-point arguments more natural:

$$\langle(\mathbf{T}_{++}[\mathcal{F}])^2\rangle_\Omega = \frac{V_{d-3}^2}{8(2\pi)^{2(d-3)}}(\delta^+)^{-d-2}(\delta^-)^{2-d}\gamma^{2d}$$
$$\times \int_0^\infty \frac{d\ell_+ d\ell'_+}{(2\pi)^2} \int_0^\infty \frac{dw\, dw'}{(2\pi)^2}(ww')^{\frac{d-4}{2}}\ell_+\ell'_+\, e^{-\mathcal{S}\left[\gamma(\ell_++\ell'_+), \gamma\left(\frac{w+1}{4\ell_+}+\frac{w'+1}{4\ell'_+}\right)\right]}. \quad (65)$$

In general, since $e^{-\mathcal{S}}$ is a just a rewriting of the smearing function, it can be somewhat arbitrary (although positive) at small arguments, however at large argument we will assume that it is damped so that the smearing function is appropriately smooth in position space. With this assumption, the general strategy is to allow $\gamma$ to be large enough such that the $\ell_+$ and $\ell'_+$ saddles of $e^{-\mathcal{S}}$ lie in this damped regime and evaluation by saddle-point is a good approximation. While this strategy should work in general, the details of it are specific to the smearing function. To be definite, let us illustrate this with a Gaussian smearing function:

$$\mathcal{F}^{\text{Gauss}}(x^+/\delta^+, x^-/\delta^-) = \frac{1}{\sqrt{2\pi}} e^{-\frac{x^{+2}}{4\delta^{+2}} - \frac{x^{-2}}{4\delta^{-2}}}, \quad (66)$$

for which we have the following two-point function:

$$\langle(\mathbf{T}_{++}[\mathcal{F}])^2\rangle_\Omega = \frac{V_{d-3}^2}{8(2\pi)^{2(d-3)}}(\delta^+)^{-(d+2)}(\delta^-)^{-(d-2)}\gamma^{2d}$$
$$\times \int_0^\infty \frac{d\ell_+ d\ell'_+}{(2\pi)^2} \int_0^\infty \frac{dw\, dw'}{(2\pi)^2}(ww')^{\frac{d-4}{2}}\ell_+\ell'_+\, e^{-\gamma^2(\ell_++\ell'_+)^2 - \gamma^2\left(\frac{w+1}{4\ell_+}+\frac{w'+1}{4\ell'_+}\right)^2}. \quad (67)$$

The $\ell_+$ and $\ell'_+$ integrals have a saddle at

$$\bar{\ell}_+ = \frac{\sqrt{w+1}}{2}, \qquad \bar{\ell}'_+ = \frac{\sqrt{w'+1}}{2}. \quad (68)$$

There is no saddle in the $w$ and $w'$ integrals and we will simply expand the exponent to linear order about their boundary value $\bar{w} = \bar{w}' = 0$. We arrive at

$$\langle(\mathbf{T}_{++}[\mathcal{F}])^2\rangle_\Omega \approx \frac{V_{d-3}^2}{2^{17/2}(2\pi)^{2d-3}}(\delta^+)^{-(d+2)}(\delta^-)^{-(d-2)}\gamma^{2(d-1)}e^{-2\gamma^2}\left(\int_0^\infty dw\, w^{\frac{d-4}{2}}\sqrt{w+1}\, e^{-\gamma^2 w}\right)^2. \quad (69)$$

The $w$ integrals can now be done analytically to yield hypergeometric functions, however given the level of approximation we have performed already it is consistent to look at the leading contribution in the large $\gamma$ limit:

$$\langle(\mathbf{T}_{++}[\mathcal{F}])^2\rangle_\Omega \approx \frac{1}{2^{11/2}(4\pi)^{d-1}}(\delta^+)^{-(d+2)}(\delta^-)^{-(d-2)}\gamma^2 e^{-2\gamma^2}. \quad (70)$$

Thus we find that at small correlation lengths, fluctuations of the DSNE are further suppressed by $e^{-2\gamma^2}$. We emphasize that this particular suppression is not universal: it is exponential because our smearing function is Gaussian. However on general grounds, we expect that for large $\gamma$ the two-point function to be suppressed by the Fourier transform of the smearing

function evaluated at a "saddle" at order $\sim \gamma$ (in dimensionless variables and up to order one factors) in its arguments:

$$\langle (\mathbf{T}_{++}[\mathcal{F}])^2 \rangle_\Omega \sim (\delta^+)^{-(d+2)}(\delta^-)^{-(d-2)} p(\gamma) |\tilde{\mathcal{G}}_{\alpha_+\gamma, \alpha_-\gamma}|^2, \tag{71}$$

where $p(\gamma)$ is a polynomial in $\gamma$ and $\alpha_\pm$ are order one constants. To recap what we have learned from this subsection and what we will take into following section:

- Vacuum fluctuations of the DSNE are finite for suitable smearing functions and scale with the smearing lengths by $\left(\delta^+\right)^{-(d+2)}\left(\delta^-\right)^{-(d-2)}$. This is multiplied by an $O(1)$ factor that is perturbative in $\gamma^2 := \delta^+\delta^- m^2$ when the correlation length is large compared to the smearing length ($\gamma^2 \ll 1$).

- When the correlation length is much smaller than the smearing length, $\gamma^2 \gg 1$, fluctuations are further suppressed by Fourier transform of the smearing function at the scale set by $\gamma$.

## 3.3 Towards a DSNEC: the squeezed states, part two

Given the boundedness of its vacuum fluctuations, our general expectation is that there is a state-independent lower bound on the DSNE and this section we will make a conjecture on its form. To motivate this conjecture we will first return to the states of section 2.2, which we remind the reader realize the UV divergence in the SNEC lower bound. For our present purposes we will express them in the Fock quantization introduced in section 3.1:

$$|\xi\rangle = \exp\left(\frac{1}{2}\int_{\mathcal{D}_{m^2}} \frac{d^2 k_\pm d^2 k'_\pm}{(2\pi)^4} \int_{S^{d-3}} \frac{d^{d-3}\Omega d^{d-3}\Omega'}{(2\pi)^{2d-6}} \left(\hat{\mathfrak{a}}^\dagger_{k_+,k_-,\Omega} \xi(k_\pm,\Omega; k'_\pm,\Omega') \hat{\mathfrak{a}}^\dagger_{k'_+,k'_-,\Omega'} - \text{h.c.}\right)\right)|\Omega\rangle, \tag{72}$$

where we will take the ansatz

$$\xi(k_\pm,\Omega; k'_\pm,\Omega') = \xi(k_+,k'_+; k_+k_-)\sqrt{k_+k'_+}(2\pi)\delta(k_+k_- - k'_+k'_-)(2\pi)^{d-3}\delta^{d-3}(\Omega+\Omega'). \tag{73}$$

The motivation for this ansatz is clear from section 2.1: the negative energy receives considerable contributions from modes with transverse momenta that are equal and anti-aligned. On-shell, $k_+k_- = \frac{\vec{p}_\perp^2 + m^2}{4}$ and so the delta functions in (73) enforce this antipodal identification of transverse momenta. We will allow $\xi$ to be general function of the magnitude, $k_+k_-$, (similar to as it was in section 2.2). The $\sqrt{k_+k'_+}$ factors are pulled out for convenience.

Using similar tricks as in section 2.2, and writing $u = 4k_+k_-$, from (56) we write the expectation value as

$$\langle \mathbf{T}_{++}[\mathcal{F}] \rangle_\xi = \frac{V_{d-3}}{2(2\pi)^{d-3}} \int_0^\infty \frac{dk_+ dk'_+}{(2\pi)^2} \int_{m^2}^\infty \frac{du}{2\pi} (u-m^2)^{\frac{d-4}{2}} \sqrt{k_+ k'_+}$$
$$\times \left( \sinh_\bullet^2(\xi)(k_+,k'_+; u/4) \mathfrak{R}\tilde{\mathcal{G}}_{\delta^+\Delta_+, \frac{\delta-u}{4k_+} - \frac{\delta-u}{4k'_+}} \right.$$
$$\left. - \sinh_\bullet(\xi) \bullet \cosh_\bullet(\xi)(k_+,k'_+; u/4) \mathfrak{R}\tilde{\mathcal{G}}_{\delta^+\Sigma_+, \frac{\delta-u}{4k_+} + \frac{\delta-u}{4k'_+}} \right), \tag{74}$$

where we recall the $\bullet$ "matrix" notation for integration over $k_+$, (41). Much like section 2.2, the key technique here is to relate (74) to an expectation value in a two-dimensional scalar theory. However, unlike section 2.2, because we are smearing over both null directions and

thus "pulling off" the light-sheet, this expectation value has to be taken in a massive theory. To be specific, let $T_{++}^{(\mu^2)}[\mathcal{F}]$ be the 2d double-null smeared stress-tensor

$$T_{++}^{(\mu^2)}[\mathcal{F}] := \int \frac{dx^+ dx^-}{\delta^+ \delta^-} \mathcal{F}(x^+/\delta^+, x^-/\delta^-)^2 : \partial_+ \hat{\varphi}(x^+, x^-) \partial_+ \hat{\varphi}(x^+, x^-) : \qquad (75)$$

of the 2d massive scalar with mass $\mu^2$

$$\hat{\varphi}(x^+, x^-) = \int_0^\infty \frac{dk_+}{2\pi\sqrt{2k_+}} \left( \hat{\alpha}_{k_+} e^{-ik_+ x^+ - i\frac{\mu^2}{4k_+} x^-} + \text{h.c.} \right). \qquad (76)$$

Note that in section 2.2, because we could restrict to $x^- = 0$ the question of whether of $\varphi$ was massive was never an issue. Let us consider its expectation value in the 2d squeezed state from section 2.2 which we reproduce here (to make explicit its $\mu^2$ dependence)

$$|\xi(\mu^2)\rangle_{2d} = \exp\left( \frac{1}{2} \int_0^\infty \frac{dk_+ dk'_+}{(2\pi)^2} \hat{\alpha}_{k_+}^\dagger \xi(k_+, k'_+; \mu^2/4) \hat{\alpha}_{k'_+}^\dagger - \text{h.c.} \right). \qquad (77)$$

For a given $\mu^2$, $\xi$ defines a 2d squeezed state in the massive theory with fixed mass, however we will be considering this in the context of an ensemble of massive theories, within which we will allow $\mu^2$ to vary. It is then simple to show that the $d$-dimensional DSNE expectation value is related by

$$\langle \mathbf{T}_{++}[\mathcal{F}] \rangle_\xi = \frac{V_{d-3}}{2(2\pi)^{(d-3)}} \int_{m^2}^\infty \frac{du}{2\pi} (u - m^2)^{\frac{d-4}{2}} \langle \xi(\mu^2) | T_{++}^{(\mu^2)}[\mathcal{F}] | \xi(\mu^2) \rangle_{2d} \Big|_{\mu^2 = u}. \qquad (78)$$

We note that (78) is essentially the generalization of equation (45) to double-null smearing. In a similar spirit to section 2.2, if we can derive a bound on the massive 2d null-energy we can apply it to lower-bound this expectation value. An additional difficulty to this, not present in section 2.2 is that this 2d theory, once pulled off the light-sheet, is no longer a CFT (as the integration in (78) is over a mass parameter) and which limits our toolbox for deriving such a bound. However it is still a free field theory and so luckily our toolbox is still plentiful. In fact, in appendix B we will show that even with $\mu^2 \neq 0$, $\langle T_{++}^{(\mu^2)}[\mathcal{F}] \rangle$ obeys a (slightly weaker[13]) "Schwarzian"-type lower bound we derived using CFT techniques:

$$\langle T_{++}^{(\mu^2)}[\mathcal{F}] \rangle \geq -\frac{1}{8\pi\delta^+\delta^-} \int d^2 x^\pm \left( \partial_+ \mathcal{F}\left( \frac{x^+}{\delta^+}, \frac{x^-}{\delta^-} \right) \right)^2. \qquad (79)$$

This bound is however not very useful in the present case: because it is insensitive to the mass, the integral over $u$ appearing in (78) is divergent. Introducing a hard UV cutoff of this transverse momenta, $\Lambda_u = 2\pi \left( \frac{d-2}{V_{d-3}} \right)^{\frac{2}{d-2}} a^{-2}$, we find

$$\langle \mathbf{T}_{++}[\mathcal{F}] \rangle_\xi \geq -\frac{1}{8\pi a^{d-2}} \int \frac{d^2 x^\pm}{\delta^+\delta^-} \left( \partial_+ \mathcal{F}\left( \frac{x^+}{\delta^+}, \frac{x^-}{\delta^-} \right) \right)^2, \qquad (80)$$

which takes the familiar SNEC form trivially integrated over the $x^-$ direction.

Happily, in appendix B we prove a more general (and more useful) family of bounds on

---

[13] Recall that the bound implied by the "Schwarzian," e.g. that appearing in (19), has a coefficient of $-\frac{1}{12\pi}$ as opposed to $-\frac{1}{8\pi}$.

$\langle T_{++}^{(\mu^2)}[\mathcal{F}]\rangle$ of which (79) is a special case. This family of bounds depend on fixed reference frames in momentum space which we label with a boost parameter, $e^\eta$:

$$\langle T_{++}^{(\mu^2)}[\mathcal{F}]\rangle \geq -\frac{\delta^+\delta^-}{4\pi}\int\frac{d^2q_\pm}{(2\pi)^2}\left|\tilde{\mathcal{F}}(\delta^+q_+,\delta^-q_-)\right|^2\Theta(q_\eta-\mu)e^{-2\eta}q_\eta\sqrt{q_\eta^2-\mu^2}, \quad (81)$$

with $q_\eta = e^\eta q_+ + e^{-\eta}q_-$. Equation (81) is true for any state $|\psi\rangle$ and any $\eta \in \mathbb{R}$. This free parameter is useful as it allows us to pull out the appropriate scaling from (81) by choosing $e^\eta = \sqrt{\frac{\delta^+}{\delta^-}}$. Substituting (81) into (78) we can perform the $u$ integration to arrive at

$$\langle \mathbf{T}_{++}[\mathcal{F}]\rangle_\xi \geq -\frac{\left(\delta^+\right)^{-\frac{d+2}{2}}\left(\delta^-\right)^{-\frac{d-2}{2}}}{4(4\pi)^{\frac{d-1}{2}}\Gamma\left(\frac{d+1}{2}\right)}\int\frac{d^2\rho_\pm}{(2\pi)^2}|\tilde{\mathcal{F}}(\rho^+,\rho^-)|^2\rho_0\left(\rho_0^2-\gamma^2\right)^{\frac{d-1}{2}}\Theta\left(\rho_0-\gamma\right), \quad (82)$$

where we recall $\gamma^2 = \delta^+\delta^-m^2$ is the dimensionless mass to smearing ratio from the previous section and we have introduced $\rho_0 = \rho_+ + \rho_- \equiv \delta^+q_+ + \delta^-q_-$. We emphasize that the integral in (82) is phrased completely in terms of dimensionless variables; we expect it to converge for smearing functions with the following large $\rho_\pm$ behaviour

$$\lim_{\rho_\pm\to\infty}|\tilde{\mathcal{F}}(\rho_+,\rho_-)|\lesssim\rho_\pm^{-\frac{d+1}{2}}, \quad (83)$$

only affecting $O(1)$ constants and not the parametric dependence on the smearing lengths, at least when the mass is small (i.e. $\gamma^2 \ll 1$). We briefly point out that (82) implies a timelike world-line bound on the null-energy of the massive scalar in $d = 4$ written down by Fewster and Roman in 2002, [29]. To see this, we can set $\delta^+ = \delta^- \equiv \delta$ imagine letting $\mathcal{F}(s^+, s^-)$ factorize as a function of $(s^0, s^1)$[14]. To be more specific, and to make contact with their notation, we will let[15]:

$$\frac{1}{\delta}\mathcal{F}\left(\frac{x^+}{\delta},\frac{x^-}{\delta}\right) := \frac{1}{\sqrt{2}}g(x^0)g_1(x^1), \quad (84)$$

for which the dependence on $g_1$ trivially integrates to one in the bound. We can then imagine letting $g_1$ being a sharply peaked Gaussian limiting to $(g_1(x^1))^2 = \delta(x_1)$. For $d = 4$ we find

$$\int dx^0\, g(x^0)^2\langle \mathbf{T}_{++}(x^0, x^1 = 0, \vec{y}_\perp = 0)\rangle_\xi \geq -\frac{(\ell_\mu v_+^\mu)^2}{12\pi^3}\int_m^\infty dq_0|\tilde{g}(q_0)|^2q_0\left(q_0^2-m^2\right)^{3/2}, \quad (85)$$

where $\tilde{g}$ is the Fourier transform of $g$, $\ell \equiv \partial_t$ is the normalized tangent vector to the timelike geodesic, and $v_+^\mu$ comes from the definition of $\mathbf{T}_{++}$, (8). One can compare this to equation (III.9) of [29].

It is also interesting to investigate (82) in the large mass scenario, $\gamma^2 \gg 1$, to see if we find the same suppression suggested by the two-point function in 3.2. To be specific we will again take $\mathcal{F} = \mathcal{F}^{\text{Gauss}}$ from equation (66) and precede by saddle-point in the $\rho_+$ integral:

$$\langle \mathbf{T}_{++}[\mathcal{F}^{\text{Gauss}}]\rangle_\xi \geq -\frac{(\delta^+)^{-\frac{d+2}{2}}(\delta^-)^{-\frac{d-2}{2}}}{2(4\pi)^{\frac{d-3}{2}}\Gamma\left(\frac{d+1}{2}\right)}\int\frac{d^2\rho_\pm}{(2\pi)^2}e^{-2\rho_+^2-2\rho_-^2}\rho_0\left(\rho_0^2-\gamma^2\right)^{\frac{d-1}{2}}\Theta(\rho_0-\gamma). \quad (86)$$

---

[14]Some care needs to be taken to be consistent with how we fixed the variance of $\mathcal{F}$ with respect to $s^\pm$ (see footnote 11), however this does not preclude such a factorization. For instance the Gaussian ansatz, (66) factorizes in the $(s^0, s^1)$ variables.

[15]The extra factors stem from matching the normalization $\int\frac{d^2x^\pm}{\delta^2}\mathcal{F}^2 = 1$ with the normalization $\int dx^0 dx^1 (g(x^0)g_1(x^1))^2 = 1$.

By writing $\rho_- = \frac{u}{\rho_+}$ we find a saddle[16] in the exponential at $\bar{\rho}_+ = \sqrt{|u|}$. Performing the fluctuation integral in $\delta\rho_+ = \rho_+ - \sqrt{u}$ about this saddle we have

$$\langle \mathbf{T}_{++}[\mathcal{F}^{\text{Gauss}}]\rangle_\xi \gtrsim -\frac{(\delta^+)^{-\frac{d+2}{2}}(\delta^-)^{-\frac{d-2}{2}}}{2^{3/2}(4\pi)^{\frac{d-2}{2}}\Gamma\left(\frac{d+1}{2}\right)} \int_0^\infty \frac{du}{2\pi} e^{-4u}\left(4u - \gamma^2\right)^{\frac{d-1}{2}} \Theta\left(4u - \gamma^2\right). \tag{87}$$

The $u$ integral is now easy to perform leaving

$$\boxed{\langle \mathbf{T}_{++}[\mathcal{F}^{\text{Gauss}}]\rangle_\xi \gtrsim -\frac{(\delta^+)^{-\frac{d+2}{2}}(\delta^-)^{-\frac{d-2}{2}}}{2^{5/2}(4\pi)^{d/2}} e^{-m^2\delta^+\delta^-}.} \tag{88}$$

This kind of suppression is reminiscent of the well-known decay of massive field correlators at space-like separations comparable with the correlation length, although we caution that the specific exponential decay arises from the choice of a Gaussian smearing function. More generally, via saddle-point arguments, we would expect the right-hand side of (88) to be suppressed by $|\tilde{\mathcal{F}}(\alpha_+ m^2\delta^+\delta^-, \alpha_- m^2\delta^+\delta^-)|^2$ where $\alpha_+$ and $\alpha_-$ are order one constants.

Lastly, we mention that this behavior has been hinted at before: in [18] it was noted that the supremum of $\langle \mathbf{T}_{\mu\nu}\rangle u^\mu u^\nu$ along a portion of a time-like geodesic, $\lambda$, of fixed smearing length $\tau_0$ (here $u^\mu$ is the normalized velocity of that geodesic) is bounded below by a quantity that is exponentially suppressed in the mass

$$\sup_{\tau\in(-\tau_0/2,\tau_0/2)} \langle \mathbf{T}_{\mu\nu}(\lambda(\tau))u^\mu u^\nu\rangle \gtrsim -k_d m^d (m\tau_0)e^{-m\tau_0}, \tag{89}$$

(where $k_d$ is a constant) which follows from optimizing world-line bounds derived in [30] for smeared weak energy, $\int d\tau \langle \mathbf{T}_{\mu\nu}(\lambda(\tau))u^\mu u^\nu\rangle g(\tau)^2$, over smearing functions.

# 4 Discussion

In this paper we investigated several new aspects of smeared null energy. While specifically the computations we performed apply for the free massive scalar, we have some expectation that our results apply broadly to Lorentzian quantum field theories for reasons we will discuss shortly below. To recap our results, we used light-sheet quantization to prove a conjectured bound on the SNE, the null energy smeared with a smooth function along a light-ray. This bound makes explicit reference to a UV cutoff, which appears in this case as a transverse "discretization" of the light-sheet. Regardless of the appearance of the UV cutoff, this "field theory SNEC" is still a potentially useful bound on its own. In the context of effective field theory, one may have a physically motivated cutoff scale (restricting the momenta of relevant field configurations) and for which a Gaussian theory with relevant interactions is a natural starting place. Additionally, although the version of SNEC proven here is not directly related to a bound in semi-classical gravity, given the general expectation that any realistic UV cutoff should be much less than the inverse Planck length (divided by the number of fields [17]), $a^{-(d-2)} \ll \mathcal{N}^{-1}\ell_P^{-(d-2)}$, one may regard our proof as a heuristic argument in favor for the existence of the gravitational form of the SNEC, as originally proposed in [1], and in a regime far from the proof of [16].

We went further on to argue that smearing over several light-rays along the same light-sheet

---

[16]There is also a saddle at $\bar{\rho}_+ = -\sqrt{|u|}$, however this does not contribute to the integral due to the Heaviside function.

does not improve the bound (in the sense of removing its cutoff dependence) and illustrated this with a series of squeezed states whose negative smeared null-energy can take values all the way up to the UV cutoff. Motivated by bounding null-energies without reference to a cutoff, we posit that the null stress tensor smeared along both null directions (that is $x^+$ and $x^-$) (what we coin the DSNE) is an operator that is bounded below for all states (at least for smooth enough smearing functions). This is motivated by showing that fluctuations over the vacuum are bounded and by evaluating the DSNE in the above series of squeezed states. The squeezed state expectation values suggest a form of the lower bound of the DSNE which we conjecture to be generally true. Importantly this conjectured bound displays a transition with the ratio of the invariant smearing length with the correlation length and for massive theories can be substantially tighter than the bound for massless theories.

The DSNEC proposed here can be seen a particular manifestation of a general expectation that energy densities are well-behaved when smeared over time-like domains. Our result is very reminiscent of time-like worldline bounds on energy densities and indeed we have shown that with an appropriate choice of smearing functions, the DSNEC implies a worldline inequality. On the other hand, we suspect that such worldline inequalities do not necessarily imply the DSNEC; at the very least it is unclear to us how to foliate a diamond in the $(x^+, x^-)$ plane with time-like worldlines without arriving at trivial lower bounds from either short worldlines at the corners, or from boosted worldlines on the null edges. On a more practical note, the DSNEC provides a cutoff insensitive regularization of the field theory SNEC along a single null geodesic, say by smearing thinly in the $x^-$ direction, which is a utility fairly distinct from the worldline inequalities.

**Proving DSNEC**

In this paper we did not undertake the harder task of proving the DSNEC, (4). We expect that a proof of this (and more specifically of the bound (82)) is obtainable using the traditional tool box of field theory techniques, at least for the current domain of free massive quantum field theory. This will be addressed in a companion paper to appear in the near future [31]. The larger task of proving some version of the DSNEC for generic interacting field theories, however, is much more difficult and likely to employ a completely different set of techniques than outlined here or in the companion paper, [31]. See the below point.

**Interactions**

Because the techniques we have employed here are special to free field theories (either through assuming conformal properties on $\mathcal{L}$ or through direct Fock quantization) it is reasonable to ask what is the fate of the SNEC or the DSNEC in interacting field theories. We pause to note the following distinction: the answer to this question is very different for relevant interactions as opposed to marginal or irrelevant interactions. In particular, as emphasized in [20], the ultra-locality of the horizon algebra and the horizon vacuum in null quantization are safe from the introduction of interactions as long as they do not alter the canonical structure of the theory on $\mathcal{L}$. At tree-level, any interaction term devoid of derivatives couplings will suffice. However at one-loop and higher, irrelevant and marginal couplings pose a real danger of spoiling the light-sheet algebra either through the necessity of derivative coupling counterterms or through (divergent) wave-function renormalization. Thus the regime of validity of our proof of SNEC is for Gaussian field theories in $d \geq 3$ dimensions with perturbative relevant interactions. It is also our expectation that (4) holds in such theories as well, at least for a suitable definition of the correlation length. Though admittedly not rigorous, our reason is the following: without

derivative couplings, the null stress tensor, $\mathbf{T}_{++}$, is functionally identical to the Gaussian theory, although its expectation values may differ from the Gaussian theory. However, by supposition, at large momenta the contribution of interactions to those expectation values will be negligible and so we expect that since (4) holds for free theories, the expectation value $\langle \mathbf{T}_{++}[\mathcal{F}] \rangle$, if negative, will still be bounded below. Dimensional analysis and the behavior of $\mathbf{T}_{++}$ under boosts suggests that this bound will still be of the form (4).

For strongly interacting field theories, we likely need a completely separate toolbox. An allegory can be drawn contrasting the techniques used in the proof of the QNEC for free field theories [9] (drawing upon the properties of free fields on a light-sheet) and those used for proving the QNEC in interacting field theories [10] (drawing upon properties of conformal field theories under modular flow). Indeed, because strongly interacting field theories can only be suitably defined via their flow to a RG fixed point, we expect formal CFT techniques to be required to fully prove some generic version of the DSNEC. At present we are not certain as to what will work and what will not,[17] although perhaps investigating CFT states prepared by stress-tensor insertions, e.g. as in [32], or investigating DSNE expectation values in holographic descriptions of negative energy states described in [33] will provide a nontrivial first check. For now we will leave

$$\langle \mathbf{T}_{++}^{(CFT)}[\mathcal{F}] \rangle_\psi \geq -\frac{\mathcal{N} C}{(\delta^+)^{\frac{d+2}{2}}(\delta^-)^{\frac{d-2}{2}}} \,, \tag{90}$$

with $C$ an $O(1)$ constant and $\mathcal{N}$ a suitable measure of the degrees of freedom, simply as a conjecture to revisit in the future.

## Acknowledgements

We would like to thank Srivatsan Balakrishnan, Mert Besken and Tom Faulkner for helpful conversations. We particularly thank Eleni Kontou for many discussions as well as collaboration on closely related work. BF is supported by ERC Consolidator Grant QUANTIVIOL, and JRF is supported by the ERC Starting Grant GenGeoHol.

## A  Appendix: An almost-basis of functions on the light-sheet

In this appendix we add some details to the "pencil" quantization for free fields on the light-sheet, $\mathcal{L} = \{x^- = 0\}$, by identifying an "almost-basis" of modes that makes the pencil decomposition natural. One main outcome of this appendix is to see explicitly that this almost-basis provides a good approximate basis for field configurations with transverse momenta much less than the inverse pencil width: $|\vec{p}_\perp| \ll a^{-1}$, establishing $a$'s role as an (inverse) UV cutoff.

Firstly we imagine overlaying the transverse $\vec{y}_\perp$ directions of $\mathcal{L}$ with a *fixed* square lattice with spacing $a$. We will label the vertices of this lattice by an integer vector $\vec{\mathfrak{p}}$. These will be the pencil labels of section 2. To each $\vec{\mathfrak{p}}$ we can associate a function on $\mathcal{L}$

$$v_{k_+,\vec{\mathfrak{p}}}(x^+, \vec{y}_\perp) := \frac{e^{-ik_+ x^+}}{\sqrt{2|k_+|}} \prod_{i=2}^{d-1} \frac{\Theta\left(1 - \frac{|y_\perp^i - \mathfrak{p}^i a|}{a/2}\right)}{\sqrt{a}} \,, \tag{91}$$

---

[17]For instance, isolating the null stress tensor from a light-cone OPE (ala [13]) will suffer additional contributions from the finite separation in the other null direction.

where $\Theta(x)$ is a Heaviside function taking the value 1 for $x \in [0, 1)$ and zero everywhere else. We will denote the transverse support of $v_{k_+,\vec{\mathfrak{p}}}$ by $D_{\vec{\mathfrak{p}}} = \left\{ \vec{y}_\perp \,\middle|\, |y^i - \mathfrak{p}^i a| < a/2 \right\}$ and the product of Heaviside functions as $\Theta_{D_{\vec{\mathfrak{p}}}}(\vec{y}_\perp)$.

It is easy to check that with respect to the Klein-Gordon norm on $\mathcal{L}$,

$$(f_1, f_2) := i \int dx^+ d^{d-2}\vec{y}_\perp \left( f_1^*(x^+, \vec{y}_\perp) \partial_+ f_2(x^+, \vec{y}_\perp) - \partial_+ f_1^*(x^+, \vec{y}_\perp) f_2(x^+, \vec{y}_\perp) \right), \quad (92)$$

that these functions are orthonormal:

$$(v_{k_+,\vec{\mathfrak{p}}}, v_{k'_+,\vec{\mathfrak{p}}'}) = \delta_{\vec{\mathfrak{p}},\vec{\mathfrak{p}}'} (2\pi) \delta(k_+ - k'_+). \quad (93)$$

Away from the lightsheet, $v_{k_+,\vec{\mathfrak{p}}}(x^+, \vec{y}_\perp)$ can be extended to a full solution, $\psi_{k_+,\vec{\mathfrak{p}}}(x^+, x^-, \vec{y}_\perp)$ to the wave equation

$$4\partial_+ \partial_- \psi_{k_+,\vec{\mathfrak{p}}} = (\nabla_\perp^2 - m^2) \psi_{k_+,\vec{\mathfrak{p}}}, \quad (94)$$

with initial condition $\psi_{k_+,\vec{\mathfrak{p}}}(x^+, x^- = 0, \vec{y}_\perp) = v_{k_+,\vec{\mathfrak{p}}}(x^+, \vec{y}_\perp)$. By writing $\psi_{k_+,\vec{\mathfrak{p}}}(x^+, x^-, \vec{y}_\perp) = = e^{ik_+ x^+} \tilde{\psi}_{k_+,\vec{\mathfrak{p}}}(x^-, \vec{y}_\perp)$, $\tilde{\psi}_{k_+,\vec{\mathfrak{p}}}$ satisfies a Schrödinger-type first order equation

$$i\partial_- \tilde{\psi}_{k_+,\vec{\mathfrak{p}}}(x^-, \vec{y}_\perp) = \frac{1}{4k_+} \left( \nabla_\perp^2 - m^2 \right) \tilde{\psi}_{k_+,\vec{\mathfrak{p}}}(x^-, \vec{y}_\perp), \quad (95)$$

which uniquely specifies it.

The obvious downside to the collection $\{v_{k_+,\vec{\mathfrak{p}}}\}$ is that they do not span the set of functions on $\mathcal{L}$ and so do not provide a full basis of field configurations. However they do provide an *almost-basis* for field configurations with long wavelengths compared to the pencil width: a field configuration with $|\vec{p}_\perp| \ll a^{-1}$ can be expanded as a combination of $\{v_{k_+,\vec{\mathfrak{p}}}\}$ up to an error on the order of $O(p_\perp^2 a^2)$, which we show now. Let $\Phi(x^+, \vec{y}_\perp)$ be a typical field configuration on $\mathcal{L}$ expressed as

$$\Phi(x^+, \vec{y}_\perp) = \int \frac{dk_+}{2\pi \sqrt{2|k_+|}} \int \frac{d^{d-2}p_\perp}{(2\pi)^{d-2}} \tilde{\phi}_{k_+,\vec{p}_\perp} e^{-ik_+ x^+ - i\vec{p}_\perp \cdot \vec{y}_\perp}, \quad (96)$$

for some $\tilde{\phi}_{k_+,\vec{p}_\perp}$. For such a $\Phi$ we define $\Phi^{(disc.)}$ as

$$\Phi^{(disc.)}(x^+, \vec{y}_\perp) := \int \frac{dk_+}{2\pi} \sum_{\vec{\mathfrak{p}}} \alpha_{k_+,\vec{\mathfrak{p}}} v_{k_+,\vec{\mathfrak{p}}}(x^+, \vec{y}), \quad (97)$$

with coefficients $\alpha_{k_+,\vec{\mathfrak{p}}}$ given by

$$\begin{aligned}
\alpha_{k_+,\vec{\mathfrak{p}}} &:= (v_{k_+,\vec{\mathfrak{p}}}, \Phi) \\
&= \int d^{d-2}y_\perp \frac{\Theta_{D_{\vec{\mathfrak{p}}}}(\vec{y}_\perp)}{a^{\frac{d-2}{2}}} \left( \int \frac{d^{d-2}p_\perp}{(2\pi)^{d-2}} e^{-i\vec{p}_\perp \cdot \vec{y}_\perp} \tilde{\phi}_{k_+,\vec{p}_\perp} \right) \\
&= a^{\frac{d-2}{2}} \int \frac{d^{d-2}\vec{p}_\perp}{(2\pi)^{d-2}} e^{-i\vec{p}_\perp \cdot (a\vec{\mathfrak{p}})} \tilde{\phi}_{k_+,\vec{p}_\perp} \mathcal{R}(\vec{p}_\perp a),
\end{aligned} \quad (98)$$

and we have defined

$$\mathcal{R}(a\vec{p}_\perp) := \prod_{i=2}^{d-1} \left( \frac{2}{ap_\perp^i} \right) \sin\left( \frac{ap_\perp^i}{2} \right). \quad (99)$$

We can think of $\alpha_{k_+,\vec{\mathfrak{p}}}$ as the coarse-graining of $\Phi$ over the pencil centered at $\vec{\mathfrak{p}}$ (after the trivial Fourier transform in $x^+$). Now consider the overlap of $\Phi$ with $\Phi^{(disc.)}$:

$$(\Phi, \Phi^{(disc.)}) = \int \frac{dk_+}{2\pi} \int \frac{d^{d-2}p_\perp}{(2\pi)^{d-2}} \frac{d^{d-2}p'_\perp}{(2\pi)^{d-2}} \tilde{\phi}^*_{k_+,\vec{p}_\perp} \tilde{\phi}_{k_+,\vec{p}'_\perp}$$
$$\times (2\pi)^{d-2} \sum_{\vec{z}\in\mathbb{Z}^{d-2}} \delta^{d-2}\left(\vec{p}_\perp - \vec{p}'_\perp - \frac{2\pi}{a}\vec{z}\right) \mathcal{R}(a\vec{p}_\perp)\mathcal{R}(a\vec{p}'_\perp), \qquad (100)$$

which also happens to be equal to $(\Phi^{(disc.)}, \Phi^{(disc.)})$. The periodic delta function comes from a sum of $a^{d-2}e^{-i(\vec{p}_\perp-\vec{p}'_\perp)\cdot(\vec{\mathfrak{p}}a)}$ over pencils and organizes the momenta in terms of the first Brillouin zone, as is familiar in lattice physics. Importantly we see that if $\tilde{\phi}_{k_+,\vec{p}_\perp}$ only has support for $|\vec{p}_\perp| \ll a^{-1}$ then the delta function can only be satisfied on the first band of this zone: $\vec{p}_\perp = \vec{p}'_\perp$. Moreover, perturbatively expanding the expression for $\mathcal{R}$ for this regime of momenta we have

$$(\Phi, \Phi^{(disc.)}) \approx \int \frac{dk_+}{2\pi} \int \frac{d^{d-2}\vec{p}_\perp}{(2\pi)^{d-2}} \tilde{\phi}^*_{k_+,\vec{p}_\perp} \tilde{\phi}_{k_+,\vec{p}_\perp} \left(1 - \frac{1}{24}a^2 p_\perp^2 + \dots\right). \qquad (101)$$

The error then in approximating $\Phi$ by $\Phi^{(disc.)}$ is small for field configurations with wavelengths much larger than the pencil width is

$$\mathcal{E}[\Phi] := (\Phi - \Phi^{(disc.)}, \Phi - \Phi^{(disc.)}) = O(a^2\vec{p}_\perp^2). \qquad (102)$$

If one is worried about the Klein-Gordon norm not being positive definite, this argument can be repeated with the $L^2$ norm to the same conclusion. Thus if we adopt $a^{-1}$ as a UV cutoff and restrict the path-integral to field configurations with transverse momenta much less than $a^{-1}$ then we can imagine always approximating field configuarations by their corresponding $\Phi^{(disc.)}$ (we will from here on drop the superscript "$(disc.)$") and quantize the theory on $\mathcal{L}$ in the pencil basis by promoting the coefficients $\alpha_{k_+,\vec{\mathfrak{p}}}$ to operators:

$$\hat{\Phi}(x^+, x^- = 0, \vec{y}_\perp \in D_{\vec{\mathfrak{p}}}) = a^{-\frac{d-2}{2}} \int_0^\infty \frac{dk_+}{2\pi\sqrt{2k_+}} \left(\hat{\alpha}_{k_+,\vec{\mathfrak{p}}}e^{-ik_+x^+} + \hat{\alpha}^\dagger_{k_+,\vec{\mathfrak{p}}}e^{ik_+x^+}\right), \qquad (103)$$

satisfying commutation relations

$$[\hat{\alpha}_{k_+,\vec{\mathfrak{p}}}, \hat{\alpha}^\dagger_{k_+,\vec{\mathfrak{p}}'}] = (2\pi)\delta(k_+ - k'_+)\delta_{\vec{\mathfrak{p}},\vec{\mathfrak{p}}'}. \qquad (104)$$

Note that $\{\hat{\alpha}_{k_+,\vec{\mathfrak{p}}}, \hat{\alpha}^\dagger_{k_+,\vec{\mathfrak{p}}}\}$ satisfy the commutation relations of decoupled 2d chiral scalars, $\{\hat{\varphi}_{\vec{\mathfrak{p}}}(x^+, x^-)\}$ quantized on $x^- = 0$. The pencil label, $\vec{\mathfrak{p}}$, acts as an internal index of the scalars. The original scalar is related to these chiral scalars via

$$\hat{\Phi}(x^+, x^- = 0, \vec{y}_\perp \in D_{\vec{\mathfrak{p}}}) = a^{-\frac{d-2}{2}} \hat{\varphi}_{\vec{\mathfrak{p}}}(x^+, x^- = 0). \qquad (105)$$

This is of course the familiar relation noted by [20]; we have simply arrived at it in way that makes the role of $a^{-1}$ as a UV cutoff manifest.

## B   Appendix: Bounds on the null energy of the 2d massive scalar

In this section we consider the null-quantization of the 2d massive scalar with mass, $\mu^2$ and its negative null-energy. In canonical quantization

$$\hat{\varphi}(x^+, x^-) = \int_0^\infty \frac{dk_+}{2\pi\sqrt{2k_+}} \left(\hat{\alpha}_{k_+}e^{-ik_+x^+ - i\frac{\mu^2}{4k_+}x^-} + \text{h.c.}\right). \qquad (106)$$

The null stress-tensor is given by $T_{++} =: \partial_+\hat{\varphi}\partial_+\hat{\varphi} :$. In the free-theory the normal ordering is unequivocally defined by Fock-mode normal ordering, but equivalently it is simply a subtraction of the bare stress-tensor, $T_{++}^{(bare)}(x^+, x^-) = \partial_+\hat{\varphi}(x^+, x^-)\partial_+\hat{\varphi}(x^+, x^-)$, of its contact divergences which are encapsulated by its vacuum expectation value:

$$T_{++} = T_{++}^{(bare)} - \langle T_{++}^{(bare)}\rangle_\Omega. \tag{107}$$

To make sense of the above expression, we will define it through point splitting,

$$T_{++}^{(bare)}(x^\pm) = \lim_{y^\pm \to x^\pm} \partial_+\varphi(x^\pm)\partial_+\varphi(y^\pm). \tag{108}$$

Now we consider smearing $T_{++}$ with $\mathcal{F}(x^\pm/\delta^\pm)^2$. With the appropriate introduction of a delta-function we have

$$T_{++}[\mathcal{F}] = \frac{1}{\delta^+\delta^-} \int d^2x^\pm d^2y^\pm \int \frac{d^2\rho_\pm}{(2\pi)^2} e^{-i\rho\cdot\Delta x} \mathcal{F}\left(\frac{x^\pm}{\delta^\pm}\right) \mathcal{F}\left(\frac{y^\pm}{\delta^\pm}\right) \\ \times \left(\partial_+\varphi(x^\pm)\partial_+\varphi(y^\pm) - \langle\partial_+\varphi(x^\pm)\partial_+\varphi(y^\pm)\rangle_\Omega\right), \tag{109}$$

where $\Delta x^\pm = x^\pm - y^\pm$. The integrand is symmetric[18] under $x^\pm \leftrightarrow y^\pm$ and so we can restrict the $\rho$ integration to an appropriate half-space: $\int \frac{d^2\rho_\pm}{(2\pi)^2} \to 2\int_H \frac{d^2\rho_\pm}{(2\pi)^2}$. The first term of (109), $\int_H d^2\rho_\pm \int d^2x^\pm d^2y^\pm \mathcal{F}_x\mathcal{F}_y T_{++}^{(bare)}$ is an inherently a positive operator (integrating the product of an operator and its Hermitian conjugate); it is the normal-ordering that is responsible for sourcing any possible negative null-energy. Thus the smeared null-energy in any state $|\psi\rangle$ is bounded below by the smeared vacuum expectation value:

$$\langle T_{++}[\mathcal{F}]\rangle_\psi \geq -\frac{2}{\delta^+\delta^-} \int d^2x^\pm d^2y^\pm \int_H \frac{d^2\rho_\pm}{(2\pi)^2} e^{-i\rho\cdot\Delta x} \mathcal{F}\left(\frac{x^\pm}{\delta^\pm}\right) \mathcal{F}\left(\frac{y^\pm}{\delta^\pm}\right) \langle\partial_+\varphi(x^\pm)\partial_+\varphi(y^\pm)\rangle_\Omega. \tag{110}$$

This statement is true for any halfspace of the $\rho^\pm$ plane, which essentially amounts to a choice of reference frame. As such, any bound derived for a specific choice of $H$ will break covariance under Lorentz boosts. In principle, however, this bound could be optimized over all possible reference frames and we expect that this minimization restores covariance. In this appendix we will take the slightly less ambitious approach and consider a family of reference frames related by a constant boost. That is we will use the momentum half-space defined by $H_\eta = \{\rho_\eta = e^\eta\rho_+ + e^{-\eta}\rho_- \geq 0\}$. We note that bound we will find depends explicitly on $\eta$. We have

$$\langle T_{++}[\mathcal{F}]\rangle_\psi \geq -\delta^+\delta^- \int_{H_\eta} \frac{d^2\rho_\pm}{(2\pi)^2} \int_0^\infty \frac{dk_+}{2\pi} k_+ \left|\tilde{\mathcal{F}}(\delta^\pm q_\pm)\right|^2\Big|_{q_+=k_++\rho_+,\, q_-=\rho_-+\frac{\mu^2}{4k_+}}, \tag{111}$$

with the constraint that $\rho_\eta = e^\eta\rho_+ + e^{-\eta}\rho_- = e^\eta q_+ + e^{-\eta}q_- - \left(e^\eta k_+ + e^{-\eta}\frac{\mu^2}{4k_+}\right) \geq 0$. We can replace the $\rho$ integrals for $q$ integrals keeping track over the appropriate integration domain:

$$\langle T_{++}[\mathcal{F}]\rangle_\psi \geq -\delta^+\delta^- \int \frac{d^2q_\pm}{(2\pi)^2} \left|\tilde{\mathcal{F}}(\delta^\pm q_\pm)\right|^2 \int_0^\infty \frac{dk_+}{2\pi} k_+ \Theta\left(q_\eta - e^\eta k_+ - e^{-\eta}\frac{\mu^2}{4k_+}\right), \tag{112}$$

where $q_\eta = e^\eta q_+ + e^{-\eta}q_-$. We perform the linear $k_+$ integral over the domain lying between $\frac{e^{-\eta}}{2}\left(q_\eta \pm \sqrt{q_\eta^2 - \mu^2}\right)$:

$$\boxed{\langle T_{++}[\mathcal{F}]\rangle_\psi \geq -\frac{\delta^+\delta^-}{4\pi} \int \frac{d^2q_\pm}{(2\pi)^2} \left|\tilde{\mathcal{F}}(\delta^\pm q_\pm)\right|^2 \Theta(q_\eta - \mu) e^{-2\eta} q_\eta \sqrt{q_\eta^2 - \mu^2}.} \tag{113}$$

---

[18]This is an obvious statement for the classical fields, but here remains true for the normal-ordered $T_{++}$ because the commutator of $\varphi$ is state-independent.

Equation (113) is true for any $\eta \in \mathbb{R}$. Let us explore some interesting limits of (113). For instance by taking the mass, $\mu^2$, and the boost parameter, $\eta$, to zero, we perform the inverse Fourier transform to find

$$\langle T_{++}[\mathcal{F}]\rangle_\psi \big|_{\mu^2=0} \geq -\frac{1}{8\pi} \int \frac{d^2 x^\pm}{\delta^+ \delta^-} \left\{ (\partial_+ + \partial_-) \mathcal{F}\left(\frac{x^+}{\delta^+}, \frac{x^-}{\delta^-}\right) \right\}^2, \tag{114}$$

consistent with the time-like bounds of [30]. Alternatively we can choose to boost this answer to a lightsheet by taking the $\eta \to \infty$ limit. The leading terms in these integrals are

$$\langle T_{++}[\mathcal{F}]\rangle_\psi \geq -\frac{\delta^+ \delta^-}{4\pi} \int \frac{d^2 q_\pm}{(2\pi)^2} \left| \tilde{\mathcal{F}}(\delta^\pm q_\pm) \right|^2 \Theta(q_+) q_+^2, \tag{115}$$

and so in the $\eta \to \infty$ limit we recover the "Schwarzian" bound, insensitive to the mass, with however a slightly weaker coefficient ($\frac{1}{8\pi}$ as opposed to $\frac{1}{12\pi}$):

$$\langle T_{++}[\mathcal{F}]\rangle_\psi \geq -\frac{1}{8\pi} \int \frac{d^2 x^\pm}{\delta^+ \delta^-} \left( \partial_+ \mathcal{F}\left(\frac{x^+}{\delta^+}, \frac{x^-}{\delta^-}\right) \right)^2. \tag{116}$$

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
