# Peer review of "Semi-local Bounds on Null Energy in QFT"

_SciPost Physics, doi:SciPost Phys. 12, 084 (2022)_

## Round 1 · Referee Report · Aron Wall · 2021-11-8

Strengths

1. Clearly written and accessible
2. Derives some new results about smeared energy conditions

Weaknesses

1. Article does not make very clear what the utility of the new bounds, are compared to past work.
2. Terminology not clearly compatible with past work on SNEC.

Report

This paper has an interesting discussion of smearing the stress tensor in null directions. As far as I can tell the results are valid, though not extremely suprising given past work on smearing the stress-tensor in [1] and [29]. The authors give a very clear explanation of their work, and citations seem to be adequate.

While this is a good paper which should be published, I'm having difficulty seeing the case for publishing in SciPost Physics given the PRL-like acceptance criteria. I don't feel that any of the 4 Expectations (at least one required) were met. The authors do not spend much time discussing the physical significance or applicability of the bound to physical problems. Since one of the bounds depends on the UV cutoff, and the other seems like a repackaging of a timelike smearing bound, it is not clear that this paper opens up a major new line of investigation.

I think this article should instead be published in SciPost Physics Core.

Requested changes

1. The authors' version of SNEC is not really the same thing as the SNEC defined in [1], since one bound uses G while the other is a flat space bound defined with a UV cutoff. Authors should consider whether making some sort of explicit terminology distinction between these bounds would be less confusing.

  • validity: high
  • significance: good
  • originality: ok
  • clarity: top
  • formatting: good
  • grammar: excellent

Author:  Jackson Fliss  on 2021-11-26  [id 1974]

(in reply to Report 1 by Aron Wall on 2021-11-08)
Category:
reply to objection

We thank Dr. Wall for his insightful comments on our manuscript and for his suggested improvements. After taking into account Dr. Wall’s comments, we would maintain that SciPost Physics is the appropriate section for publishing this manuscript, for the following reasons:

(1) In this manuscript we prove a version of the SNEC directly from field theory and valid in a large class of theories. While acknowledging that this bound makes explicit use of a UV cutoff, it is still a relevant and potentially useful bound for effective field theories (which have a physically motivated cutoff scale and are often relevant deformations of Gaussian theories). Additionally, assuming that a reasonable UV cutoff should be much less than the inverse Planck length, our proof gives strong credence to the original SNEC proposal in a regime where it would not be directly verifiable, say, via holography. Given the SNEC’s established utility in semi-classical gravity (e.g. [Freivogel, Kontou, Krommydas; 2020]) we feel that this result is a significant addition to the literature. With that in mind, we are grateful for Dr. Wall’s suggestion to better distinguish the quantity we prove from the original proposed SNEC, to avoid equivocation. In our revision we will take care in our Introduction to clearly state this distinction and to explain the relation to the SNEC as originally proposed, as well as clarify the points above.

(2) We would like to emphasize that the DSNEC proposed in this paper is logically distinct from time-like worldline inequalities. While one can derive worldline inequalities from the DSNEC (as illustrated in the manuscript), it is not clear that one can derive the DSNEC from foliating a diamond in the $(x^0,x^1)$ plane with worldline inequalities, say, due to the pinching of the worldlines at the corners. We acknowledge a pervading understanding that quantities smeared over time-like domains have finite lower bounds, however we believe that our result manifests this understanding in a way previously unexplored in the literature. An additional novelty to the above is the motivation of the DSNEC as a way to regulating bounds along a null geodesics (such as the field theory SNEC) in a cutoff-independent manner: here the regulator appears directly from thinly smearing in the other null direction as opposed to appearing from a UV sensitive cutoff. Of course, our conjectured bound is more general than the above situation, however this motivation is logically distinct, say, from smearing the null energy along a time-like worldline. We plan to make the novelty of this result mentioned above clearer in our revision.

We hope that Dr. Wall will find our comments and a soon-to-be-submitted revision satisfactory.

-J.R. Fliss, B. Freivogel

---

## Round 2 · Author Response

After taking into account the suggestions and comments from the referee, we are submitting a minor revision that implements the points that we brought up in our response. A detailed list of the changes can be found below.

We hope that this submission better clarifies the novelty and utility of our results as well appropriately distinguishes our SNEC result from its original context, as requested by the referee.

-J.R. Fliss, B. Freivogel

---

## Round 2 · List of Changes

1) Added remarks to paragraph below equation (3) in the Introduction to distinguish the SNEC result from the SNEC originally proposed in the context of semi-classical gravity. Added remark in the same paragraph on how this result gives credence to the original SNEC and added a reference.

2) Expanded the first paragraph of the Discussion to remark on the utility of the SNEC result in the context of effective field theories and to elaborate on the relation of the result to the original SNEC.

3) Added paragraph to Discussion (third paragraph) emphasizing the novelty of the proposed DSNEC compared to known worldline inequalities.

---

## Editorial Decision

published